# Some Thermoelectric Phenomena in Copper Chalcogenides Replaced by Lithium and Sodium Alkaline Metals

**DOI:** 10.3390/nano11092238

**Published:** 2021-08-30

**Authors:** Marzhan M. Kubenova, Kairat A. Kuterbekov, Malik K. Balapanov, Rais K. Ishembetov, Asset M. Kabyshev, Kenzhebatyr Z. Bekmyrza

**Affiliations:** 1Faculty of Physics and Technical Sciences, L.N. Gumilyov Eurasian National University, Nur-Sultan 010008, Kazakhstan; kkuterbekov@gmail.com (K.A.K.); assetenu@gmail.com (A.M.K.); kbekmyrza@yandex.kz (K.Z.B.); 2Physical and Technical Institute, Bashkir State University, 450076 Ufa, Russia; balapanovmk@mail.ru (M.K.B.); ishembetovrkh@rambler.ru (R.K.I.)

**Keywords:** thermoelectric materials, copper sulfide, crystal structure, conductivity, diffusion, thermal conductivity, Seebeck coefficient, superionic conductors

## Abstract

This review presents thermoelectric phenomena in copper chalcogenides substituted with sodium and lithium alkali metals. The results for other modern thermoelectric materials are presented for comparison. The results of the study of the crystal structure and phase transitions in the ternary systems Na-Cu-S and Li-Cu-S are presented. The main synthesis methods of nanocrystalline copper chalcogenides and its alloys are presented, as well as electrical, thermodynamic, thermal, and thermoelectric properties and practical application. The features of mixed electron–ionic conductors are discussed. In particular, in semiconductor superionic copper chalcogenides, the presence of a “liquid-like phase” inside a “solid” lattice interferes with the normal propagation of phonons; therefore, superionic copper chalcogenides have low lattice thermal conductivity, and this is a favorable factor for the formation of high thermoelectric efficiency in them.

## 1. Introduction

In recent years, mixed electron–ion conductors, and in particular, semiconductor superionic copper selenide and sulfide, have become the objects of intensive research by scientists involved in the development and study of thermoelectric materials (TE) due to the “discovery” of their “liquid-like state” of ions, which reduces the lattice thermal conductivity of the crystal to record low values [1,2]. The development of nanotechnology has made the possibilities of modifying materials to improve their useful properties virtually limitless. For example, the addition of even a small fraction of nanosized particles makes the bulk material nanocomposite and noticeably improves its thermoelectric characteristics. This does not complicate the technology of its production [3,4].

The heat-to-electricity energy conversion efficiency of an ideal thermoelectric generator is determined by the Carnot efficiency and materials’ performance as [4]:(1)$$\eta =\left(\frac{T_{hot}-T_{cold}}{T_{hot}}\right)\left[\frac{\sqrt{1+ZT_{avg}}-1}{\sqrt{1+ZT_{avg}}+\left(\frac{T_{cold}}{T_{hot}}\right)}\right],$$
where *T_hot_* is the hot-side temperature and *T_cold_* is the cold-side temperature; *ZT*_avg_ is the average dimensionless thermoelectric figure-of-merit, which is a critical measure for materials’ performance. *ZT* value of material is calculated from the formula
*ZT* = *α*^2^*σT*/*k*,(2)
where *α*, *σ* and *k* are the Seebeck coefficient, electrical conductivity and thermal conductivity of a material, correspondingly.

Achieving the optimal combination of all three material properties at the same time is a challenge. In addition, for the practical application of thermoelectric material, the manufacturability of production, the availability and cheapness of raw materials, stability of properties, mechanical resistance, environmental friendliness of production, and other factors are important. Currently, among industrially produced materials, the most widespread is doped bismuth telluride (Bi_1−*x*_Sb*_x_*)_2_(Se_1−y_Te_y_)_3_, which has a figure of merit *ZТ* ≈ 1 at room temperature. Despite the fact that many materials have been obtained under laboratory conditions that are superior in thermoelectric characteristics to bismuth telluride, for the reasons stated above, it has remained the most demanded commercial thermoelectric material for several decades [3,4].

The excellent thermoelectric properties of copper and silver chalcogenides have long been known [5,6,7], but their practical application is hindered by the high rate of copper diffusion. At elevated temperatures, rapid degradation of thermoelements occurs due to copper release. For this reason, in the 1980s, the developments of American physicists on the use of silver-doped copper selenide in thermoelectric elements [8] were curtailed; similar problems are described in [9,10]. The boom of interest of specialists in this class of materials arose recently after the publication of the article [1] by a group of Chinese researchers with the group of G.J. Snyder (USA). It focused on the superionic, “liquid-like” crystalline nature of copper selenide, which helps to reduce the lattice thermal conductivity [11]. This and subsequent publications actually created a promising new direction—the design of effective thermoelectric materials through a decrease in thermal conductivity, by creating conditions that suppress the propagation of phonons, but do not impede electronic transport [2]. The classification of thermoelectric materials includes the concept of “superionic thermoelectric materials”. In addition, to increase *ZТ*, a targeted change in the synthesis conditions for the modification of known materials and nanostructuring are often used today [2,3,4,12,13,14,15].

Over the past 3–4 years of intensive research, the figure of merit *ZT* has been significantly increased: for copper selenide to *ZT* = 2.1 at 700 °C [16], for copper sulfide Cu_1.97_S to *ZT* = 1.9 at 700 °C [17]. However, the problem of these materials remains the risk of rapid degradation of the material [18], which remarkably reduces the practical significance of the above-mentioned works.

Active studies of the anomalously fast diffusion of ions in solids began with the development of ideas about the defect structure (ideas of Schottky, Frenkel, and K. Wagner [19,20]). It was found that fast diffusion in superionics is due to the peculiarities of their crystal structure. Moreover, high ionic conductivity is observed in crystals with strong structural disorder. For example, in silver iodide, iodine ions form a rigid lattice skeleton, and silver ions constantly move through the numerous voids of the lattice. The degree of disorder in the silver sublattice is higher than in liquid, and when the crystal melts, the rate of silver diffusion decreases. The fact of correlation between the degree of lattice disordering and the value of ionic conductivity is described, for example, in [20,21].

Studying copper sulfides and copper selenides substituted with lithium, it was found that the ionic conductivity and diffusion rate of copper in materials decrease by almost an order of magnitude compared to the initial binary compositions, and the thermoelectric coefficient remains high and even increases [22,23,24,25,26,27,28,29]. This significantly improves the prospects of copper chalcogenides for practical use. In 2017, the work of a large international group of researchers (USA, Canada, China) [30] was published, in which, in continuation of works [24,25], they studied copper selenide doped with lithium. The lithium was reacted with a fine powder of Cu_1.9_Se by mechanical alloying using a ball miller for 90 min under Ar. To consolidate the powder into a pellet, the powder was hot-pressed at 45 MPa and 700 °C for 1 h under an Ar atmosphere using a high-density graphite die in an induction furnace. The work of Kang, S.D. et al. [30] confirmed that doping with lithium improves the stability of copper selenide while maintaining high thermoelectric characteristics. For the composition Li_0.09_Cu_1.9_Se, they obtained the maximum *ZT* ≈ 1.4 at 727 °C. In a recent work by Ge, Z.H. et al. [31], the thermoelectric properties of copper sulfide Cu_1.8_S (digenite) doped with sodium are described. Doping with sodium and nanostructuring allowed them to increase the figure of merit *ZT* of copper sulfide from 0.6 to 1.1 at 500 °C.

Superionic thermoelectric materials have some peculiarities; for example, in [32,33,34,35], a sharp abrupt increase in the value of *ZT* was observed in the region of the superionic phase transition. This fact deserves special attention and further detailed research, since it gives hope to use it as another way to improve the thermoelectric characteristics. In the chalcogenides, the region of the superionic phase transition can occupy several tens of degrees, and the transition onset temperature depends on nonstoichiometry and the presence of an impurity that stabilizes the superionic state at a lower temperature [19]. Lowering the phase transition temperature can be used to transfer the working area of the thermoelectric to the region of lower temperatures, in which copper diffusion is significantly reduced while maintaining a high *ZT* value, which should greatly reduce material degradation.

In this respect, lithium-substituted copper sulfides and selenides look more suitable, since they can be single-phase even at room temperature [25]. Sodium-substituted sulfides, with a noticeable sodium content, are mixtures of various sulfide phases at room temperature [36,37,38,39,40], or individual chemical substances are formed with properties far from properties of initial thermoelectric material; for example, NaCu_4_S_4_ [41] represents the behavior of an ideal metal. However, multiphase sodium-containing copper sulfides obtained in the nanocrystalline state turn out to be nanocomposites, which favorably affects their thermoelectric and thermal properties.

In the past 10–15 years, the main trend in improving thermoelectric efficiency is an inhibition of a thermal conductivity of a material [2,3,4,42]. Most of the ways to reduce thermal conductivity while maintaining high electrical conductivity of the material are somehow related to the nanostructuring of the material. According to a review article on nanostructured thermoelectrics by P. Pichanusakorn and P. Bandaru [42], while total nano-object materials such as superlattices and nanowires do promise a significant decrease in lattice thermal conductivity *k*_L_*—*mainly due to a decrease in mean free path *λ*, problems in large-scale and reliable synthesis in fact preclude their widespread use, so bulk materials with embedded nanoscale elements now look more suitable for practical applications. In our opinion, this conclusion is still relevant, and nanocomposites and compacted mixtures of nanocrystalline materials require more attention from researchers in order to better understand their properties and learn how to control them to improve their thermoelectric properties.

The general goal of this review is to consider copper chalcogenides and their alloys as thermoelectric materials from the point of view of the researcher of the superionic state, since it is this state that provides these materials with beautiful thermoelectric properties and excellently low thermal conductivity. In addition, we wanted to show convenient electrochemical methods for working with copper and silver chalcogenides, which are not in demand by the current generation of researchers. These ideas developed in the works of C. Wagner, I. Yokota, and S. Miyatani; for example, the coulometric titration method and methods for measuring transport characteristics in direct dependence on Fermi level are very suitable for tuning thermoelectric power by changing the copper content. However, it is the superionic state that complicates the practical application of copper and silver chalcogenides, since the mobility of cations similar to the mobility of atoms in a liquid leads to the release of the metal from the sample during a long operation at high temperatures. Therefore, we also considered here the possibilities of overcoming this important problem: associated with a decreasing in ionic conductivity of the material and with the design of thermoelectric modules, developed taking into account the threshold of metal release from chalcogenide.

In addition, we consider here some other TE materials, such as half-Heusler phases, skutterudites, etc., so that the reader has the opportunity to compare and can more fully evaluate the superionic copper chalcogenides as thermoelectric materials.

## 2. Materials

The host thermoelectric materials that are the objects of our review, namely, copper chalcogenides Cu_2−*x*_X (X = S, Se), have both high electronic conductivity and high ionic conductivity. They are called mixed conductors and also superionic semiconductors. Their high electronic conductivity is caused by the presence of a large number of vacancies in the cation sublattice, which determines the p-type conductivity. They have a wide region of homogeneity over the copper sublattice; that is, they are phases of variable composition, and retain the type of crystal lattice when the stoichiometric index of copper changes from 2 to 1.75. This makes it possible to control the electrical, thermoelectric, diffusion, and other properties of these compounds by changing the degree of nonstoichiometric composition, which tunes concentration of electron holes and carrier’s effective mass, too.

The high cationic conductivity of copper chalcogenides is possible due to the action of several factors, and the main ones are the strong disorder of the cation sublattice in the superionic phase and the presence of a connected network of free interstitial sites, which are shallow potential wells in the path of mobile ions. The value of the ionic conductivity in the superionic phase of Cu_2_Se reaches 2.2 S/cm at 400 °C, and the self-diffusion coefficient equals 2.5 × 10^−5^ cm^2^/s at 350 °C [22]. Due to the high concentration of free interstitial sites in the cation sublattice, the activation energy for copper diffusion does not include the energy of defect formation and consists only of the activation energy for migration of cations. For copper selenide, the activation energy of ionic conductivity is 0.15 ± 0.01 eV [22], and for hexagonal superionic phase of copper sulfide, it is 0.19 ± 0.02 eV [24].

### 2.1. Crystal Structure and Phase Transitions

#### 2.1.1. Cu-S System

Copper and sulfur form a wide variety of compounds, ranging from chalcocite (Cu_2_S) to villamaninite (CuS_2_) with other intermediate phases: djurleite (Cu_1.96_S), roxbyite (Cu_1.8125_S), digenite (Cu_1.8_S), anilite (Cu_1.75_S), geerite (Cu_1.6_S), spionkopite (Cu_1.39_S), yarrowite (Cu_1.12_S), and covellite (CuS) [43,44]. There are still controversial points on the crystal structure of low-temperature sulfide phases, but the discussion of these disagreements is not included in the subject of our review; therefore, we will only briefly present information from the latest structural works on these materials.

Chalcocite *γ*-Cu_2_S below 104 °C is described by the monoclinic space group P21/c with a unit cell containing 48 Cu_2_S formula units [45]. Between 104 °C and 435 °C, *β*-Cu_2_S chalcocite is hexagonal with the space group *P6_3_/mmc*. Above 435 °C, high-temperature chalcocite *α-*Cu_2_S transforms into a cubic close-packed structure of digenite $Fm\bar{3}m$ [19,43]. At room temperature, chalcocite usually exists as a mixture with djurleite, since the two phases readily transform into each other. Djurleite (Cu_1.965_S ÷ Cu_1.934_S) has a monoclinic lattice (space group *P21/n*) and is stable up to 93 ± 2 °C [45], then reversibly decomposes into hexagonal chalcocite Cu_1.988_S and hexagonal digenite Cu_1.84_S [46].

Digenite (Cu_1.8_S) exists in two forms: a low-temperature phase (below 91 °C) and a high-temperature cubic form with the space group $Fm\bar{3}m$ (above 91 °C) [47]. According to Roseboom [46], the copper content in digenite increases with temperature and reaches the composition Cu_2_S at 435 °С. Below 72 °C, digenite forms a metastable phase, which transforms into orthorhombic anilite [43]. The high-temperature cubic modification contains four copper atoms (ions) with uncertain coordination in the unit cell. The equilibrium distribution of copper atoms over the voids of the “framework” of the lattice can be calculated based on the principle of the maximum configuration entropy. In [48], the indexing of X-ray reflections and the distribution of cations in Cu_1.8_S at room temperature was carried out according to this principle. It was found that copper ions preferably occupy tetrahedral positions (1/4, 1/4, 1/4) (within the zinc blende structure) and trigonal positions (1/3, 1/3, 1/3). Octahedral positions remain vacant. The influence of temperature and the degree of deviation from stoichiometry on the coordination of mobile cations in various polymorphic modifications is currently insufficiently reflected in the available literature. At the same time, the system of mobile cations certainly affects the thermodynamics and kinetics of polymorphic transformations and the semiconducting properties of copper sulfide [20].

Anilite (Cu_1.75_S) is relatively stable and forms at a temperature of 75 ± 3 °C [43]. Anilite occurs naturally as a mineral. Its crystal structure is rhombic with cell parameters a = 7.89 Å, b = 7.84 Å, c = 11.01 Å. Sulfur atoms form a rigid skeleton of the lattice, while copper atoms are ordered in interstices. The ordered distribution of copper atoms leads to a slight displacement of sulfur atoms from the nearest cubic positions. With a sulfur content of more than 36.36% at room temperature, anilite coexists with covellite (CuS) [43]. Covellite (CuS) is hexagonal with *P6_3_/mmc* space group [18,43], and it does not exist as a single phase at room temperature [43]. Potter [49] showed that covellite is stoichiometric within 0.0005 of the *n(Cu)/n(S)* mole ratio. When anilite is heated above 75 ± 2 °C, a mixture of cubic digenite and covellite is formed [43]. The results of Dennler et al. [18] showed that CuS is not stable at temperatures above 180 °C either in air or in an N_2_ atmosphere, and the material was observed to decompose to Cu_2_S and S. It is in accordance with the earlier paper of D. Shah [50], which concluded decomposition of CuS to Cu_2_S and S in air.

Roxbyite (Cu_1.8125_S), according to Mumme, W.G. et al. [50], has a triclinic lattice, with space group $P\bar{1}$, with unit cell dimensions a = 13.4051 (9) Å, b = 13.4090 (8) Å, c = 15.4852 (3) Å, *α* = 90.022 (2), *β* = 90.021 (2), *γ* = 90.020 (3) [51]. The structure of roxbyite is based on a hexagonal close-packed framework of sulfur atoms with the copper atoms occupying these layers, all having triangular coordination. Other layers sandwiched between the close-packed sulfur layers consist purely of double or split layers of Cu atoms. Some of these Cu atoms have twofold linear coordination, but mostly they have three- and fourfold coordination to the sulfur atoms in the close-packed layers that lie above and below them. The crystal structure of roxbyite bears a strong kinship to those of low chalcocite and djurleite [51].

The crystal structures of minerals spionkopite (Cu_1_._39_S) and yarrowite (Cu_1.12_S) were first described by Goble [52] in 1980. Yarrowite has a hexagonal lattice with a = 3.800 (1) Å, c = 67.26 (4) Å, *Z* = 3. Spionkopite also is hexagonal with a = 22.962 (3) Å, c = 41.429 (1) Å, *Z* = 18. Space groups for both minerals are *P3ml*, $P\bar{3}m1$, or *P321*. Yarrowite and spionkopite have well-developed subcells that strongly resemble the unit cell of covelline.

#### 2.1.2. Cu-Na-S System

##### Solid Solutions on Base of Copper Sulfide

The wide range of homogeneity of superionic copper sulfide along the metal sublattice (from Cu_2_S to Cu_1.75_S) allows doping with other metals while maintaining the type of crystal structure, allowing one to obtain homogeneous samples with the desired useful properties.

The effect of sodium doping on transport phenomena in copper sulfide was investigated by Z. H. Ge et al. [31]. The thermoelectric properties of bulk samples of copper sulfide Na*_x_*Cu_1.8_S (*x* = 0, 0.005, 0.01, 0.03, 0.05), consolidated using the technology of spark plasma sintering from a nanopowder with an average nanoparticle size of 3 nm, synthesized by mechanical fusion, are described. The limit of sodium solubility in the crystal structure of the sulfide is revealed as *x* = 0.01. The purpose of the doping was to reduce the conductivity and increase the Seebeck coefficient. According to measurements of the Hall effect, in the samples doped with sodium, the concentration of carriers decreases, compared to pure Cu_1.8_S. In addition, the presence of many nanosized pores and grains was found, which led to a decrease in thermal conductivity by a factor of 2–3. As a result, a high value of *ZT* = 1.1 at 500 °С for Na_0.01_Cu_1.8_S was achieved in this work, mainly due to a decrease in thermal conductivity, which is almost twice as high as for pure Cu_1.8_S, and is comparable in magnitude with modified PbS materials (*ZT* ≈ 1.2 at 650 °C). The sodium solubility in the interstices of the Cu_1.8_S lattice is 0.28%; at a higher sodium concentration (for the Na_0.05_Cu_1.8_S alloy), inclusions of the Na_2_S and Cu_1.96_S phases were observed.

Note that light sodium doping was recently studied by Z. Zhu et al. [53] for copper selenide. Doping with Na by mechanical fusion introduces multiple micropores, which can optimize heat transfer through strong phonon scattering at the interfaces between micropores and grains. The introduction of micropores is an effective way to improve thermoelectric characteristics, which is similar to another effective method of introducing a secondary nanophase [3,16,42]. According to the results of our dissertation work, it is the presence of additional nanoscale phases that explains the high thermoelectric characteristics of the sulfide alloys studied in the work.

##### Binary Compounds of Na-S System

The properties of Na_2_S are well studied. In general, it has a crystalline structure of fluorite and a cubic modification, ionic conductivity (~0.1 S cm^−1^ at 723 °C) is high even for superionic conductors, and weakly depends on cationic substitution, which indicates a high disorder (“melting”) of the cationic sublattice. The energy of motion was found by B. Bertheville et al. [54] to be 0.61 ± 0.05 eV for a cation vacancy. As is known, the lattice parameter is 6.5373 Å, the degree of filling of the cation sublattice is 0.988 [55], the strong disordering of the structure provides a low lattice component of thermal conductivity. In these systems, the properties of the transition of an electron occurs without a change in the momentum of an electron; the difference in the energies of electrons between the bottom of the conduction band and the top of the valence band is 2.23–3.05 eV [55,56]. In [57], by M. Kizilyalli et al., it is reported that new Na_2_S structures of cubic and rhombic symmetry were obtained at high temperatures. The approximate unit cell parameters were found to be a = 11.29 Å for the cubic form and a = 15.94 Å, b = 16.00 Å, and c = 16.18 Å for the orthorhombic form.

##### Na_2_Cu_4_S_3_

Na_2_Cu_4_S_3_ phase was found by Savelsberg G. and Schäfer H. [58,59]. Na_2_Cu_4_S_3_ has a monoclinic lattice type, with the space group *C2/m*. The lattice parameters are a = 1563 (3), b = 386 (2), c = 1033 (2) *pm*, *β* = 107.6°. These atoms together form sulfur atoms in layers, which are stacked in the direction of the c-axis through separated octahedrally coordinated Na atoms. After that, all copper atoms are triple coordinated with sulfur. In addition to the two sulfur atoms, there are six copper neighbors, as sulfur atoms have layers surrounded by three copper atoms. Burschka C. and Naturforsch Z. [60] described the crystal structure of Na_3_Cu_4_S_4_, which also belongs to the thiocuprate class.

##### NaCu_5_S_3_

The crystal structure of NaCu_5_S_3_ was studied by H. Effenberger et al. [61]. The ternary system was obtained by the hydrothermal synthesis method. NaCu_5_S_3_ is hexagonal with space group *P 6322-D 6*, Z = 2 [61]. The structural parameters are a = 6.978(5) Å, c = 7.209(6) Å. The formation energy of the system is 0.382 eV. NaCu_5_S_3_ decomposes into Cu_1.75_S + Na_3_(CuS)_4_ + Cu. The S atom has an irregular coordination figure created by two neighboring Na atoms and four copper atoms. W. Yong et al. [62] studied the optical properties of the NaCu_5_S_3_ thiocuprate. The Cu-S structure doped with Na alkali metal ions was synthesized by the hydrothermal method. The diffuse reflectance spectrum shows that the band gap of Cu_2_S nanocrystals is 1.21 eV; after doping with Na^+^, the color of the product strongly changed. The optical band gap measured at the edge of the absorption band of NaCu_5_S_3_ was 0.49 eV, which indicates a decrease in the photoelectric properties in the visible region of the spectrum.

##### NaCu_4_S_4_

Zhang X. et al. [63] studied the compound NaCu_4_S_4_. The structure of NaCu_4_S_4_ is reported to consist of a two-dimensional Cu/S framework of trigonal symmetry. The compounds are carried out through Cu-S bonds with the participation of metal atoms from a layer of the GaS type of sulfur atoms. If all monosulfides and disulfides have a charge of *2^−^*, then the charges on the metal decrease to Na(Cu^+^)_2_(Cu^2+^)(S_2_)S_2_. However, the chalcogen present in ternary systems, in addition to copper, is also in a mixed valence state. This situation is similar to the situation in CuS, where the formal charge of Cu is *1^+^* and the average charge of S is *1^−^*. Thus, NaCu_4_S_4_ represents the behavior of an ideal metal. In NaCu_4_S_4_, the [CuS] framework has a 0.25^−^ total charge, and the average charge S decreases even more to 1.25^−^, still short 2^−^ for the filled S^2−^ *p*-band. The addition of extra electrons to the sulfur [CuS] bands results in a less localized state with a significant degree of delocalization.

In the works of Klepp K. et al. [64,65], the crystal structure of Na_4_Cu_2_S_3_ [64] and Na_7_Cu_12_S_10_ [65] thiocuprates with discrete anions was investigated. Na_4_Cu_2_S_3_ is tetragonal, with the space group *I4_1_/a* with a = 9.468(1) Å, c = 36.64(2) Å and *Z* = 16. An outstanding feature of Na_4_Cu_2_S_8_ is the formation of discrete V-shaped thiocuprate anions [S-Cu-S-Cu-S]^4−^ with copper in an almost linear coordination by sulfur. The bond length (d_Сu–S_ = 2.15 Å) and angles are in good agreement with the infinite anionic chains of KCuS [66]. The existence of a compound with NaCuS composition was not confirmed.

#### 2.1.3. Cu-Li-S System

The ternary system Cu-Li-S has been poorly studied. Only Cu_2_S, Li_2_S, and LiCuS compounds were extensively investigated. It was shown in the works of Balapanov et al. [24,25] that in the Li_x_Cu_2−*x*_S system, solid solutions are formed based on the superionic f.c.c. phase of copper sulfide Cu_2_S up to *x* = 0.25. At a lithium content above *x* ≈ 0.15, a solid solution is formed already at room temperature; in the range 0 < *x* <0.15 at room temperature, a mixture of phases is observed, which, upon heating, gradually transforms into a solid solution based on the cubic phase of copper sulfide with the space group $Fm\bar{3}m$. The crystal structure and phase relationships at a lithium content above *x* = 0.25 have not been studied. In the Cu-Li-Se system, there is information about the crystal structure and phase transitions for the composition Li_0.25_Cu_1.75_Se [23]. In contrast to ternary sulfide, ternary selenide exhibits a more complex pattern of phase transitions, and phase transition to f.c.c. structure shifts above 500 °C, while Li_0.25_Cu_1.75_S structure is cubic at room temperature [25]. In the paper of Balapanov et al. [25], by neutron diffraction studies, it is shown that gradual disordering in the cation sublattice with the temperature increasing leads to changes in the symmetry of the crystal lattice of the superionic conductor Li_0.25_Cu_1.75_Se at 127 and 227 °C. At temperatures close to these values, anomalies in the temperature dependence of both ionic and electron conductivity are observed. Let us consider in more detail the available information on the crystal structure and phase transitions in the Li-Cu-S system.

##### Li_2_S

Lithium sulfide has the antifluorite structure at ambient conditions, space group $Fm\bar{3}m$, *Z* = 4 [67] with cell parameter a = 5.7158(1) Å. It undergoes a diffuse (“Faraday”) phase transition to a fast ion conduction region at about 527 °C and is referred to as a superionic conductor [68,69]. Its high ionic conductivity (≈0.15 S/cm at 727 °C) as a consequence of a Frenkel defect formation without any significant distortion of the f.c.c. sulfur sublattice. The Li diffusion process is carried out by hopping between regular tetrahedral and interstitial octahedral sites. The mean residence times on the regular Li sites were estimated to be 17.3 *ps* at 900 °C, 6.7 *ps* at 1000 °C, and 4.3 *ps* at 1090 °C [69]. The phase transition at 527 °C is also confirmed by studies of Brillouin scattering [70].

Elastic neutron diffraction of Li_2_S, measured as a function of temperature in [68], shows the onset of a diffuse phase transition near 627 °C to a superionic state. Inelastic neutron scattering has been used to investigate the harmonic lattice dynamics of Li_2_S at 15 K. The authors of [68] conclude that the present data set suggests that in Li_2_S, the simple defect structure, the occupation of (½, ½, ½) sites, is created. A shell model has been successfully fitted to the data. The results of the band structure and electron density in Li_2_S are presented in the paper of Tsuji J. [71]. From the results of calculations of the electronic structure, it was found that Li_2_S is a semiconductor with an indirect band gap, while a similar Na_2_S compound is a semiconductor with a direct band gap.

##### Solid Solutions on Base of Copper Sulfide

Judging by the similarity of the crystal structure and the closeness of the lattice parameters of the binary compounds Li_2_S and Cu_2_S, as well as the closeness of the ionic radii of Cu^+^ and Li^+^ ions, it can be assumed that they can form solid solutions, at least at high temperatures. In similar systems with heavier alkaline cations (Cu_2_S-K_2_S, Cu_2_S-Rb_2_S, Cu_2_S-Tl_2_S), incommensurate quasi-one-dimensional structures of the ACu_7_S_4_ type (A = Tl, K, Rb) [72,73] are formed, which are chemically equivalent to the Li_0.25_Cu_1.75_S compound (which, however, has a cubic structure). In the range from −243 °C to 127 °C, six to seven phase transitions with superstructures are observed in these compounds. Their properties are explained using the theory of charge density waves. Such quasi-one-dimensional structures are not synthesized with sodium, but other crystal types are realized as thiocuprates. Apparently, such structures are energetically unfavorable with light cations of alkali metals. A similar situation is observed for structures of the type [74] with a monoclinic *C12/m1* lattice—there are homologous compounds, but futile attempts to obtain an isostructural compound with sodium [75,76]. It can be stated that at a large value of the ratio of the radii of the chalcogen/metal ions, other crystal structures are more favorable.

The authors of [24,25,26,27,28,29] experimentally investigated the crystal structure and transport properties of Li*_x_*Cu_2−*x*_S solid solutions, which exhibit superionic conductivity. X-ray diffraction studies [25] revealed that Li*_x_*Cu_2−*x*_S (*х ≤* 0.25) compounds are solid solutions on the base of α-Cu_2_S at temperature higher than certain temperature, which depends on the chemical composition of the phase. According to the work of Balapanov et al. [25], the presence of lithium in the lattice in a sufficiently high concentration (0.10 < *x* < 0.25) significantly reduces the temperature of the phase transition in Li*_x_*Cu_2−*x*_S to the cubic $Fm\bar{3}m$ phase. Lithium ions in the lattice preferably occupy octahedral 32*(f)_II_* positions, while copper ions are mostly distributed in tetrahedral 32*(f)_I_* positions. The authors of [25] conclude that since the “easy diffusion paths” of copper ions pass through the octahedral voids, this reduces the ionic conductivity of Li*_x_*Cu_2−*x*_S (*х ≤* 0.25) solid solutions compared to binary copper sulfide.

In the range of 20–500 °C, four phase transitions are observed in Li_0.25_Cu_1.75_Se by Balapanov et al. [23]. The phase transformation (PT) in Li_0.25_Cu_1.75_Se occurs at temperature (130 ÷ 140) °C from triclinic to monoclinic syngony. At 230 ÷ 242 °C, the monoclinic phase is followed by the rhombohedral modification. Both of these PTs are accompanied by drops on the calorimetric curve. At about 380 °C, observed anomalies in temperature dependencies of the ionic conductivity of the chemical diffusion coefficient and jump of ionic Seebeck coefficient have been induced by the PT from the rhombohedral to hexagonal phase of Li_0.25_Cu_1.75_Se. Neutron diffraction studies revealed the cubic structure of the Li_0.25_Cu_1.75_Se compound (with space group $Fm\bar{3}m$) at 500 °C. Copper ions are statistically distributed over tetrahedral and trigonal voids of rigid Se sublattice, and lithium ions randomly occupy *32(f)* positions.

A detailed study of the thermal and electrochemical behavior of the Cu_4−*x*_Li*_x_*S_2_ phase (*x* = 1, 2, 3) was carried out by Chen E.M. and Poudeu P. [77]. In this work, it was shown that Cu_3_LiS_2_ (*x* = 1) and LiCuS (*x* = 2) crystallize with unique crystal structures of low symmetry at room temperature. While the XRD pattern at room temperature of the sample with *x* = 3 is comparable to that of the cubic structure of Li_2_S, additional peaks observed on the XRD patterns of samples with *x* = 1 and *x* = 2 suggest lower symmetry structures for both phases near room temperature (Figure 1). Taking into account differential scanning calorimetry (DSC) measurements, the authors determined structural phase transition at 140 °C for samples with *x* = 1 and *x* = 2 from a low-symmetry rhombohedral modification to the high-temperature cubic modification of the binary compound. In addition, for this compound, a tendency of a decrease in thermal conductivity with an increase in temperature and an increase in the Cu:Li ratio was revealed, which is an essential factor for improving these systems for thermoelectric purposes.

In a paper by S.D. Kang et al. [30], the phase transition in Cu_2−*x*_Se doped lithium is discussed. The authors note that significant changes are happening with Li doping. In the parent compound Cu_2−*x*_Se, the superionic transition happens across a temperature range 103–137 °C for *x* = 0.01, in which the low temperature phase (complete structure remains unknown by the authors) and the high temperature phase (cubic, $Fm\bar{3}m$) are mixed. The transition shows strong hysteresis even at the slowest practical ramping rates, showing behavior characteristic of a first-order transition. The biggest change upon substituting Cu with Li is the splitting of the phase transition into two transitions, involving an additional intermediate phase. The high-temperature superionic phase is reached at a higher temperature of around 227 °C upon heating. The total transformation enthalpy (i.e., integrated area of the peaks) remains large, indicating that the cation sublattice melting nature of the transition still persists with Li doping. The distribution of the transformation enthalpy over a wide range of temperatures in both peaks likely indicates that both transitions happen gradually through a phase mixture region in the phase diagram. Powder X-ray diffraction also shows the existence of the intermediate phase. At 152–177 °C, when the peak characteristic of the low temperature phase is almost diminished, some peaks that are not present in the high temperature phase persist. These peaks can be indexed together with the other strong peaks using a monoclinic unit cell (a = 6.379 Å; b = 5.815 Å; c = 6.155 Å; *β* = 97.91*°*, which is similar to what has been suggested from an earlier study on Li_0.25_Cu_1.75_Se by Bickulova et al. [78] and Balapanov et al. [23]. The compound LiCuS has been studied quite well.

##### LiCuS

Kieven D. et al. [79] in 2011 studied sputtered LiCuS films with thickness ~200 nm on quartz glass for use in solar cells. In the work, both optical transmission and reflection spectra were obtained. For an indirect and a forbidden direct type of transition, the band gap E_g_ was found as 2.0 ± 0.1 eV and 2.5 ± 0.1 eV, respectively. Later, the work of German scientists from the Max Planck Institute, A. Beleanu et al. [80], determined the crystal structure of LiCuS ternary sulfide. The crystal structure was determined using neutron and X-ray powder diffraction and solid-state NMR analysis on the ^7^Li nucleus [80]. Polycrystalline Li_1.1_Cu_0.9_S was obtained by the reaction of Li foil (99.999%) with CuS powder: CuS + 1.1 Li = Li_1.1_Cu_0.9_S + 0.1 Cu. The compound crystallizes in the orthorhombic structure of Na_3_AgO_2_ type with space group *Ibam*. The crystal structure of Li_1.1_Cu_0.9_S can be obtained from the cubic structure of Li_2_S by moving a part of Li along the *c* axis, and the Li atoms become linearly coordinated by S atoms. There are 24 atoms per unit cell, occupying four different crystallographic sites. S occupies *8j* (S), and the remaining metals, Li and Cu, occupy *8g* (M1), 4c (M2), and *4b* (M3). All the metals sites are occupied by randomly mixed Li and Cu atoms. The lattice parameters decrease almost linearly with increasing lithium concentration, which is explained by the smaller ionic radius of lithium compared to the radius of copper. Performing in the work [80] the density functional theory calculations show that Li_1.1_Cu_0.9_S is a direct band-gap semiconductor with an energy gap of 1.95 eV, which is consistent with experimental data.

In the work of the Egyptian author S. Soliman [81], theoretical calculations of the band gap (1.7 eV) for half-Heusler LiCuS compound by the Engel and Vosko method are presented. Calculations were performed for each structure distribution to determine the lowest energy structure. The electronic structure calculations were performed using the lowest energy distribution as follows. The calculations were based on a spatial symmetry arrangement according to space group 216 ($F\bar{4}3m$), where Li, Cu, and S occupied positions *4a*, *4b*, and *4d*, respectively. The lattice parameter a_calc_ = 5.53 Å. The calculations showed that the distribution of the atoms for LiCuS among the above-mentioned Wyckoff positions yielded the lowest energy per formula unit for the compound.

Thus, studying the current state of research on the crystal structure of superionic copper chalcogenides, it can be stated that ions are characterized by diffuse phase transitions occupying a wide temperature range as a rule. With increasing temperature, there is a continuous redistribution of mobile copper ions over different types of interstices in the anionic core of the crystal lattice. Mobile cations can be likened to a “cationic liquid” filling the voids of the structure. The presence of a “liquid-like phase” inside a “solid” lattice interferes with the normal propagation of phonons; therefore, superionic copper chalcogenides have low lattice thermal conductivity, which is a favorable factor for the formation of their high thermoelectric efficiency. As shown below, the insufficient stability of these compounds due to the high diffusion rate of copper can be increased by introducing impurity atoms or creating a composite structure by including nano-objects, such as carbon nanotubes, graphene fragments, etc., into the superionic matrix.

### 2.2. Methods for Synthesis of Perspective Thermoelectric Materials

At present, much attention is paid to the development and study of new highly efficient thermoelectric materials in the scientific world, which can be judged by the greatly increased number of publications in this field in the past decade. Modern strategies for the search and synthesis of promising thermoelectrics are based on the quantum theory of solids and the latest advances in nanotechnology [3,4,82,83]. Depending on the group of materials (film, bulk, superlattice, supramolecular, etc.) and on the temperature range, the achieved record *ZT* values lie in range from 1 to 4. Our review is mainly devoted to bulk thermoelectric materials, including composite materials. Their dimensionless thermoelectric efficiency *ZT* does not exceed 3, as can be seen in Figure 2, which shows some of the best achievements of recent years. We have presented here mainly chalcogenide materials.

This section will briefly review synthesis methods and design techniques for bulk thermoelectric materials that have been frequently used over the past ten years.

The chemistry of the synthesis of semiconductor chalcogenides has developed very intensively in the past two decades. Priority was given to various methods of “cold” synthesis, which do not require large time and energy costs, and allow one to immediately obtain synthesis products in nanosized form.

M.R. Gao [93] describes more than 15 liquid-phase methods for the synthesis and modification of chalcogenide nanomaterials: liquid exfoliation method [94], hot-injection method [95], single-source precursor method [96,97], hydrothermal method [98], solvothermal method [99], mixed solvent method [100], microwave method [101], sonochemical method [102], electrodeposition method [103], electrospinning method [104], photochemical method [105], template-directed method [106], ion exchange reactions, and others.

The hydrothermal method has been widely used for the synthesis of a variety of functional nanomaterials with specific sizes and shapes [93]. The main advantages of hydrothermal and solvothermal processes are fast reaction kinetics, short processing times, phase purity, high crystallinity, low costs, and so on. According Gao et al. [93], the hydrothermal method has achieved great success in the preparation of nanocrystalline chalcogenides with various nanostructures, such as *β*-In_2_S_3_ nanoflowers, CuS micro-tubules, Sb_2_S_3_, Sb_2_Se_3_ and Sb_2_Te_3_ nanobelts, Ag_2_Te nanotubes, Ag_2_Se nanoparticles, and so on. In a hydrothermal process, the presence of a small quantity of organic ligands often plays a key role to determine the sizes, shapes, and structures of the nanocrystals. By carefully adjusting the *pH* value, monomer concentration, as well as the reaction temperature and reaction time, bismuth chalcogenides with various nanostructures such as nanostring-cluster hierarchical Bi_2_Te_3_ and Bi_2_S_3_ nanoribbons and Bi_2_Te_3_ nanoplates have been successfully synthesized using the hydrothermal methods. For instance, in Zhu et al.’s [53] work, the nanocrystalline Na_0.04_Cu_1.96_Se bulk samples were synthesized with excellent *ZT* = 2.1 at 700 °C by combining hydrothermal synthesis and hot pressing. By the solvothermal method, lots of MC nanocrystals with an elegant control of the size and shape distributions and also the crystallinity have been synthesized, including wire-like Cu_2_Te, Ag_2_Te, and Bi_2_S_3_; belt-like Bi_2_S_3_; flower-like γ-In_2_Se_3_; dendrite-like Cu_2-x_Se and Cu_2_S; and so on. The hot-injection method is very effective in synthesizing high-quality nanocrystals with good crystallinity and narrow size distributions [95].

Zhao Y. et al. [107] described three chemical synthesis methods for the synthesis of Cu_2-x_S (*x* = 1, 0.2, 0.03) nanocrystals (NC): sonoelectrochemical, hydrothermal, and dry thermolysis. The control of the chemical composition of NC was carried out by regulating the reduction potential in the sonoelectrochemical method, by regulating the PH value in the hydrothermal method, and by selecting the pre-treatment of precursors in the dry thermolysis method. The use of the electrochemical method of doping with lithium allowed the authors to obtain and study solid solutions Li*_x_*Cu_2−*x*_S (0 < *x* < 0.25) [24,26,28], Li*_x_*Cu_(2−*x*)−*δ*_Se (*x* ≤ 0.25) [22,23], promising for thermoelectric applications. To obtain Li*_x_*Cu_(2−*x*)−*δ*_Se nanopowders, Ishembetov R. Kh. et al. [108] used the method of electrohydrodynamic impact, which makes it possible to effectively grind even the hardest materials down to a few nanometers.

Despite the popularity of liquid-phase synthesis methods, high-temperature ampoule synthesis from elements followed by ball milling and hot pressing is also a popular method for producing high-performance thermoelectrics based on copper chalcogenides. D. Yang et al. [109] analyzed recent works on the synthesis of binary copper and silver chalcogenides. They note that for thermoelectric materials research, dense pelletized samples are required. The Cu_2−*x*_X (X = S, Se, Te) powders synthesized by the above-mentioned methods are compacted by hot pressing and spark plasma sintering (SPS). However, the temperature and electrical field involved may drive the mobile Cu ions [110], causing composition inhomogeneity and undesired microstructures in nonstoichiometric of Cu_2−*x*_X compounds. He et al. found that high packing density of Cu_2_Te samples could be obtained by pressureless direct annealing without hot pressing or SPS, and the resulting density values were comparable to those prepared by SPS [110].

For synthesis of other types of thermoelectric materials, similar methods are used. The authors of the works of D. Kraemer et al. [90] synthesized a p-type MgAgSb alloy operating at temperatures from 20 °C to 245 °C, with very high conversion efficiency of 8.5%, using a simple one-step hot pressing technology. The thermoelectric material powder was prepared by ball milling procedure.

Among many thermoelectric materials, half-Heusler compounds (space group *F43m, C1b*)—principally with the composition XYZ, where X and Y are transition or rare earth elements and Z is a main group element are the most studied ones [111]. Localized *3d* states of transition elements such as Fe, Co, Ni, etc., make the maximum of the valence band or the minimum of the conduction band flat and heavy. Thus, higher carrier concentrations, which require a higher dopant content, are required to optimize power factors. Their physical properties are largely determined by the valence electron count (VEC) [82]. High thermoelectric performance is generally achieved in the semiconducting half-Heusler phases with *VEC* = 18, while *VEC* > 18 usually leads to metallic conduction behavior [4]. Half-Heusler compounds possess several promising features, namely, a high Seebeck coefficient, moderate electrical resistivity, and good thermal stability. According to S.J. Poon [111], the RNiSn-type half-Heusler compounds, where R represents refractory metals Hf, Zr, and Ti, are the most studied to date. Since 2013, the verifiable *ZT* of half-Heusler compounds has risen from 1 to near 1.5 for both *n-* and p-type compounds in the temperature range of 500–900 °C [111]. TaFeSb-based half-Heusler phase with record high *ZT* of ~1.52 at 700 °C was prepared in 2019 by H. Zhu et al. [88] using two-step ball-milling and hot-pressing methods. In the work of H.T. Zhu et al. [112], the ZrCoBi alloy with a high *ZT* of ~1.42 at 700 °C and a high thermoelectric conversion efficiency of ~9% at the temperature difference of ~500  °C was obtained. In pair with ZrCoBi, the high-performance half-Heusler thermoelectric modules СoSb_0.8_Sn_0.2_ were used, which ensure self-propagating synthesis and optimization of the topological structure. Due to the nonequilibrium reaction process in n*-*type and p-type materials, dense dislocation matrices were introduced, which greatly reduced the lattice thermal conductivity. In the work of Austrian scientists G. Rogl et al. [113], synthesized half-Heusler compounds of the n- and p-type (Ti_0.5_Zr_0.5_ based on NiSn and NbFeSb) were obtained by the high-pressure torsion method to improve their thermoelectric characteristics due to a sharp decrease in the direction of ultralow thermal conductivity. This decrease is due to grain refinement and a high concentration of defects caused by deformation, that is, vacancies and dislocations, which are determined by severe plastic deformation.

One of the directions in the search for highly efficient thermoelectric materials remains skutterudites [83]. Skutterudites are a kind of cobalt arsenide minerals consisting of variable traces of iron or nickel substituting for cobalt to formulate CoAs_3_. The chemical formula of skutterudites can be expressed as ReM_4_X_12_, where Re is a rare earth element, M is a transition metal element, and X is a non-metal element from Group V, such as phosphorus, antimony, or arsenic. These are lead- and tellurium-free thermoelectric materials that are highly efficient for intermediate temperature range applications. These materials have the figure of merits *ZT* values close to 1.3 for p-type and 1.8 for n-type, with good mechanical stabilities [114]. For CoSb skutterudites, obtained by spark plasma sintering (SPS) at 650 °C under a pressure of 50 MPa, the thermoelectric figure of merit *ZT* ~1.7 at 577 °C has been achieved [115]. Liu et al. [116] reviewed the recent progress made in CoSb_3_-based materials and synergistic optimization of the thermal and electrical properties. Multi-filled skutterudites demonstrated inferior thermal conductivities, resulting in a considerable increase in *ZT* values [117].

Thus, by controlling the synthesis and forming nanocrystalline, nanostructured, and nanocomposite materials, it is possible to obtain a wide range of required properties of semiconducting chalcogenides for a variety of applications, as evidenced by numerous recent works [4,82,83,109].

## 3. Transport Phenomena in Mixed Electron–Ion Conductors

### 3.1. Electrical Properties of Copper Sulfide and Its Alloys

The reviewed materials on the base of copper chalcogenides as a rule have mixed electronic–ionic conductivity. However, the number of ion transfer usually does not exceed a few percent, and the determining mechanism of electrical conductivity is electron transfer. Copper chalcogenides have a fairly wide range of homogeneity over the cationic sublattice. The lack of cations in the lattice, according to the electroneutrality rule, leads to the formation of electron holes in the valence band and to p-type conductivity. Nonstoichiometric defects in Cu_2−*δ*_X (X = S, Se, Te) compounds, the concentration of which is determined by the nonstoichiometric index *δ*, play the role of an alloying component (impurity). This impurity forms shallow levels in band gap, which are usually completely ionized at room temperature and above temperatures.

In Cu_2−*δ*_X (X = S, Se, Te) compounds, usually the concentration of “impurity” holes in the valence band (n_p_) is much higher than the concentrations of uncontrolled impurities and equilibrium point defects. The concentration n_i_ of intrinsic carriers is determined by the temperature and the band gap. For nonstoichiometric compositions, at temperatures far from fusion temperature, n_i_ is also significantly less than n_p_. In this case, the temperature dependence of the electronic (hole) conductivity is determined by the temperature dependence of the mobility and has a metallic character. In compensated materials, for example, in Li*_x_*Cu_2−*x*_X (X = S, Se, Te), the temperature dependence of conductivity is semiconducting [19,20].

The peculiarities of the crystal structure and the presence of a mobile ionic subsystem create a number of problems in the interpretation of electronic kinetic effects in superionic semiconductors. Based on the work of V.M. Berezin [20], the following main aspects of these problems can be distinguished:(a)The disordered crystal structure of the studied chalcogenides leads to the fact that besides the background of the periodic potential of the anionic sublattice, charge carriers are exposed to the fluctuation potential of the cation sublattice; thus, the description of the phenomena of the transfer of electrons and holes faces the same problems as in non-crystalline solids and liquids [118]. The specificity lies in the description of the total effect of two sublattices—periodic and disordered, on electrons and holes. In this case, (polaron) effects associated with the localization of electronic wave functions (Anderson transition) become possible in the electronic system [119]. The existence of similar localized states as applied to intercalate chalcogenides with a two-dimensional character of conductivity was studied, for example, in the works of A.N. Titov [120] and Yarmoshenko Y.M. [121].(b)The ease of “overflow” of cations over the voids of the anion framework with a change in temperature or a change in the nonstoichiometry of the composition leads to a smearing of phase transitions and a continuous change in the parameters of the band structure during this redistribution (change in the effective mass of carriers, width of the gap), etc.(c)The anharmonicity of vibrations of atoms of the crystal lattice and high coefficients of self-diffusion in a disordered sublattice call into question the applicability of the developed theory of scattering of current carriers in semiconductors in the harmonic approximation. The temperature dependences of the electron and hole mobilities must be refined experimentally and new approaches to their theoretical description must be sought.

To describe the electronic energy spectrum in superionic semiconductors, it is convenient to use the functions of the density of electronic states *g(ε)*, where *ε* is energy of electron. This is a universal function of the electronic system, the use of which is not related with the periodicity and defectiveness of the atomic crystal structure. For an arbitrary isotropic dispersion law *ε(p)*:(3)$$g\left(\varepsilon \right)=\frac{8\pi }{h^{3}}p^{2}V\frac{dp}{d\varepsilon },$$
where *V* is the electronic system volume [20]. For the case of electrons in the conduction band with a parabolic dispersion law, the density of electronic states is determined as:(4)$$g\left(\varepsilon \right)=\frac{8\sqrt{2}}{h^{3}}\pi Vm^{*}{}^{\frac{3}{2}}{\left(\varepsilon -\varepsilon_{c}\right)}^{1/2},$$
where *m^*^* is effective mass of electron.

Since the dispersion law *p(ε)* in disordered systems cannot be correctly introduced, it is practically impossible to use Formula (4) to calculate *g(ε)* in the materials under study. However, in [20], another method was indicated for introducing the density of states, which can be used for practical calculations in superionic chalcogenides. Let there be a degenerate electron system with the Fermi energy ε_F_. A change in the number of electrons in this system by *ΔN* should correspond to a change in the position of the Fermi level by *Δε_F_*, which, taking into account two spin orientations, can be represented as:(5)$$\Delta \varepsilon_{F}=\frac{1}{2}\frac{\Delta N}{g\left(\varepsilon \right)}.$$

Expression (5) allows us to find the density of states near the Fermi level. As shown by Wagner [122], the change in the chemical potential *μ_p_* (Fermi level) of electron holes in copper chalcogenides is related by the formula *Δμ_e_* ≡ *Δε_F_* = *e**ΔE* to the electromotive force (e.m.f.) *E* of an electrochemical cell of type:(6)$$\mathrm{Cu}/\mathrm{CuBr}/{\mathrm{Cu}}_{2-x}S/Pt,$$
where Cu is reversible metallic electrode, CuBr is electronic filter (material with unipolar Cu^+^—ionic conductivity in range of 340–440 °C [123]), Cu_2-*x*_S is a sample, and *Pt* is inert metallic electrode. At temperatures above room temperature, all nonstoichiometric defects in nonstoichiometric copper chalcogenides are already ionized (which is often confirmed by the metallic type of temperature dependence of the conductivity); therefore, the change in the number of electron charge carriers can be expressed through the change in the nonstoichiometric index *δ*: (7)$$\Delta N={\Delta \delta N}_{A}\frac{V}{V_{m}},$$
where *V* is the sample volume, *V_m_* is the molar volume, and *N_A_* is the Avogadro number. The Formula (7) can be used for the experimental determination of change in the density of electronic states:(8)$$g\left(\varepsilon \right)=\frac{\Delta \delta \cdot V\cdot N_{A}}{2e\cdot \Delta E\cdot V_{m}},$$

In the presence of two types of carriers in a semiconductor, its conductivity *σ* is described by the equation: (9)$$\sigma =e\left(n_{e}\mu_{n}+n_{p}\mu_{p}\right),$$
where *n_e_* and *μ_n_* are the concentration and mobility of electrons, respectively, *n_p_* and *μ_p_* are the same for holes, and *e* is the electron charge.

Taking into account the relationship between the hole concentration and the nonstoichiometry degree *δ* in such sample as ${\mathrm{Cu}}_{2-\delta }S$, for example: *n_p_* = *δN_A_*/*V_m_*(10)
where *V_m_* is the molar volume of the phase and *N_A_* is Avogadro’s number. From Equation (7), one can obtain a formula to estimate the hole mobility from compositional dependence of the conductivity *σ(δ):*(11)$$\mu_{p}=V_{m}/\left[F\left(d\sigma /d\delta \right)\right],$$
where *F* = *eN_A_* = 96,480 C/mol is the Faraday number.

#### 3.1.1. Electronic Conductivity

The simplest way to study the properties of a semiconductor is to measure its electrical conductivity. As is known, the result of this measurement depends on the number of mobile charge carriers (electrons, holes), on the distribution of their thermal velocities, and on the deviation from the equilibrium distribution, which is caused by the applied electric field. The classical transport theory used is usually based on the approximation of the Boltzmann kinetic equation, which arose from the kinetic theory of gases, where the electrical conductivity is considered to be a free electron gas flow.

The Drude model, related to the drift mobility and conductivity of carriers, is discussed in connection with its applicability to explain electron transport in solids, some ideas that still remain an integral part of the classical theory of free electrons.

The usual expression for electrical conductivity is:(12)σ=neµ,
where quantity $µ=e\tau_{m}/m$ is the drift mobility of electrons.

Electrical conductivity is also often expressed in terms of the average mean free path of electrons $\lambda_{m}=v_{T}\tau_{m}$, defined as the distance traveled by an electron moving with thermal velocity $v_{T}$ during the mean free path $\tau_{m}$. Thus, in the Drude model, the electrical conductivity can be written as:(13)$$\sigma =\frac{{ne}^{2}\tau_{m}}{m}=\frac{{ne}^{2}\lambda_{m}}{mv_{T}}=\frac{{ne}^{2}\lambda_{m}}{{\left(3{mk}_{B}T\right)}^{1/2}},$$
where *k_B_* is the Boltzmann constant.

The Lorentz model demonstrates the transfer of an electric charge associated with particles of an electron gas (electrical conductivity), as well as with the transfer of kinetic energy by the same electrons (electronic thermal conductivity) [124,125]. The resulting formula for electrical conductivity:(14)$$\sigma =\frac{4{ne}^{2}\lambda_{m}}{3{\left(2\pi mk_{B}T\right)}^{1/2}}$$

From the point of view of the quantum mechanical concept, only those electrons that are near the Fermi level (Sommerfeld’s model) contribute to the electrical conductivity. These states drift in space due to the external electric field at a high speed, approximately equal to the Fermi velocity (*V_F_*), and only the movement of these electrons in the direction of the electric field can contribute to the conductivity. Thus, conductivity can be described:(15)$$\sigma =\frac{1}{3}e^{2}V_{F}^{2}\tau N\left(\varepsilon_{F}\right)$$

This quantum mechanical equation shows that the conductivity depends on the Fermi velocity (*V_F_*), the relaxation time (*τ*), and the degree of filling *N(ε_F_*) of the Fermi level, which is proportional to the density of states.

The Seebeck coefficient in the generate regime varies with the carrier concentration according to the formula (parabolic band, energy-independent scattering approximation):(16)$$\alpha =\frac{8\pi^{2}\;k_{B}^{2}}{3{eh}^{2}}m^{*}T{\left(\frac{\pi }{3n}\right)}^{2/3},$$
where *m** is the effective mass of carrier [120].

One of the first studies of the electronic conductivity of copper sulfides was the work of J.B. Wagner and C. Wagner [122]. Using the relation of Fermi level in Cu_2-__δ_S with e.m.f. *E* of the cell (I), it was found in [122] that the ratio of the effective mass of electron holes to the mass of a free electron is 7 ± 2 at 435 °C in Cu_1.8_S. Measurements of the conductivity indicate a significant increase in the mobility of electron holes with an increase in the copper deficit in the samples. In the work of I. Yokota [125] in 1953, for the example of Cu_2-δ_S, his diffusion theory of the transfer of ions and electrons in mixed conductors was presented. With the help of this theory, Yokota determined the threshold of the current density at which copper begins to separate from the Cu_2-δ_S sample as:(17)$$a\equiv \frac{ejL}{2k_{B}T\sigma_{p}}>1,$$
where *j* is a current density, *L* is a sample length, and *σ_p_* is the hole conductivity. Taking this knowledge into account is important for creating conditions for long-term operation of thermoelectric devices based on copper chalcogenides and similar materials.

Japanese scientists T. Ishikawa and Sh. Miyatani [126] in 1977 investigated the electronic and ionic conductivity and the Hall coefficient of binary copper chalcogenides depending on the electromotive force (e.m.f.) of the electrochemical cell (I) $\mathrm{Cu}/\mathrm{CuBr}/{\mathrm{Cu}}_{2-x}S/Pt$. It was established earlier by C. Wagner [122] that e.m.f. of cell of type (I) is determined by the expression: (18)$$E=\frac{\mu_{\mathrm{Cu}-}\mu_{\mathrm{Cu}}^{0}}{F},$$
where $\mu_{\mathrm{Cu}-}\mu_{\mathrm{Cu}}^{0}$ is the difference in chemical potentials of copper atom in the Cu_2-x_S sample and in metal copper electrode, correspondingly. In a steady state, the difference in chemical potentials is compensated by the resulting difference of electrical potentials, which is expressed by Equation (8). Equation (8) shows that cell e.m.f. *E* can be judged on the change in the chemical potential of copper atoms in the sample, and, consequently, on the copper content in the phase of variable composition. T. Ishikawa and Sh. Miyatani considered *E* = Δ(en)/Δε hh as the relative height of the Fermi level in copper chalcogenides. There is $\varepsilon =eE/k_{B}T$, where *E* is electron energy.

A significant increase in the mobility of electron holes at 100 °C ~2 cm^2^/V s and an effective mass of holes ~2.3 *m_0_* (where *m_0_* is the mass of a free electron) were found. The authors explain the degree of Cu excess as a result of the screening action of carriers (holes) on the vacancies of copper ions.

I. Yokota and Sh. Miyatani [127] in 1981 presented the phenomenological theory of ion–electronic conductivity of Cu_2_S based on cross-conduction and ambipolar diffusion of Cu_2_S. The cross-conductivities *σ_ie_* and *σ_ei_* for steady states at only electronic current passing and for only ionic current passing through specimen were measured by Sh. Miyatani for the *β-*phase of Cu_2_S at 340 °C; it was found that *σ_ei_*
*≈ σ_ie_* and the inverse Onsager relation is held for Cu_2_S, and the cross-conductivity values were at least 100 times lower than the ionic conductivity and the electronic conductivity.

Gafurov I.G. [128] studied electrical properties of Cu_2-x_S doped with lithium. He established that substitution by lithium leads to decreasing electronic conductivity as well as to decreasing ionic conductivity. The Seebeck coefficient of Cu_(2−*x*)−_*_δ_*Li*_x_*S (0 < *x* < 0.25) increases with a rise in nonstoichiometricity degree *δ* in the temperature interval 20 °C to 410 °C. Measured Hall mobilities of holes lie in the range of 6 ÷ 60 cm^2^ V^−1^ s^−1^. Debae temperatures *θ_D_* of superionic cubic phase of Cu_(2−*x*)−_*_δ_*Li*_x_*S are lower than *θ_D_* = 145 K for Cu_2_S; for example, *θ_D_* = 101 K for Cu_1.75_Li_0.25_S.

Ishembetov et al. [108] in 2011 studied the effect of grain size on the electronic conductivity of copper selenide Cu_1.9_Li_0.10_Se. With a decrease in the grain size to 50 nm, a decrease in electrical conductivity by a factor of 2–3 was observed. Interestingly, at temperatures above 540 °C, the Seebeck coefficient of a nanostructured sample begins to decrease, which may be associated with the development of intrinsic conductivity. Kang et al. [30] in 2017 found that doping with lithium increases the conductivity and, as a result, improves the thermoelectric performance of copper selenide. They reported a maximum *ZT* > 1.4 for Li_0.09_Cu_1.9_Se composition, prepared by solid phase alloying, ball milling, and hot pressing. Copper sulfide and selenide are very similar in their electrical properties, so something similar should occur in the Cu_2_S system. In the work [129] by M. Guan et al., the Cu_2−*x*_Li*_x_*S compounds (*х* = 0, 0.005, 0.010, 0.050, and 0.100) were studied. When *x* < 0.05, the Cu_2−*x*_Li*_x_*S samples are stable and pure phases, having the same monoclinic structure as the pristine Cu_2_S at room temperature. The electrical and thermal conductivities were measured in the temperature range from 27 °C to 627 °C. The electrical conductivity in the Cu_2−*x*_Li*_x_*S is greatly improved with the Li doping content increasing due to the enhanced carrier concentrations. For Cu_1.95_Li_0.05_S, the *σ* increases to about 87 S cm^−1^ at 27 °C, about one order of magnitude higher than that of the Cu_2_S matrix. Since doping with Li in Cu_2_S increases the ion activation energy [24] and thereby lessens the influence of mobile ions on heat-carrying phonons, it leads to a significant increase in the thermal conductivity of Li-doped Cu_2_S samples. The maximum value *ZT* = 0.84 was obtained at 627 °C for the composition Cu_1.99_Li_0.01_S, an improvement of about 133% compared to the Cu_2_S matrix. The impurity of lithium significantly reduces the conductivity both by compensating for holes and by introducing additional scattering centers. Impurity atoms also reduce the thermal conductivity of the material. In our opinion, further improvement in the thermoelectric figure of merit of lithium-doped copper chalcogenides can be achieved with a higher degree of doping with lithium and a lower copper content than in the works of Kang et al. [30] and Guan et al. [129]. Then, the cation sublattice as a whole will contain a sufficient number of vacancies to maintain high conductivity. In other words, the material must be an uncompensated semiconductor so that its conductivity is not too low. Perhaps, nonstoichiometric chalcogenides rather than Cu_1.95_S and Cu_1.90_Se should be chosen for Li doping to receive homogeneous compositions, such as Cu_1.85_S and Cu_1.85_Se.

The transfer phenomena in solid solutions of Cu_2_Se-Ag_2_Se, Cu_2_Se-Li_2_Se, and Cu_2_S-Li_2_S systems, depending on temperature, chemical composition, and nonstoichiometry degree, were studied by R.Kh. Ishembetov [130]. In the work, the temperature dependences of electronic conductivity were obtained for solid solutions of the following compositions: Ag_0.23_Cu_1.75_Se, Ag_0.5_Cu_1.5_Se, Ag_1.2_Cu_0.8_Se, and Li_0.1_Cu_1.75_Se. It is shown that substitution with silver significantly changes the electrical properties of copper selenide. While the composition Ag_0.23_Cu_1.75_Se exhibits a temperature dependence of conductivity of highly degenerate semiconductor, Ag_1.2_Cu_0.8_Se has a semiconducting character of the conductivity. The presence of an isovalent impurity (silver or lithium) leads to the appearance of a metal–semiconductor transition with temperature variation; for example, about 340 °C for the Li_0.1_Cu_1.75_Se composition, the monotonic decrease in conductivity with temperature increasing is changed to semiconductor behavior of conductivity, and for the Ag_0.5_Cu_1.5_Se composition, at the same temperature, the semiconductor character of the dependence is changed to metallic. The dependence of the electronic conductivity σ_e_ on the silver content in the solid solutions Ag_1.2±δ_Cu_0.8_Se, Ag_0.23±δ_Cu_1.75_Se, and Ag_0.5-δ_Cu_1.5_Se and on the copper content in the solid solution Li_0.1_Cu_1.75+δ_Se has been studied. For the Ag_1.2_Cu_0.8_Se composition, the dependence of the electronic conductivity on the degree of nonstoichiometry has a minimum. In the author’s opinion, the observed minimum corresponds to the stoichiometric composition, which is explained by a decrease in the concentration of defects in the cation sublattice when the stoichiometric composition is approached. With the same composition, a change in the sign of the electronic Seebeck coefficient is observed, and the hole conductivity is replaced by electronic one. It proves that the Ag_1.2_Cu_0.8_Se solid solution, in contrast to copper selenide, which exists only with a copper deficiency, can exist both with a deficiency and with an excess of metal relative to the stoichiometric composition. From the slope of the *σ_e_(δ)* dependences, the mobilities of charge carriers in the *α*-Ag_1.2±__δ_Cu_0.8_Se solid solutions at 400 °C are determined in paper [131]. For analysis, the *σ_e_(δ)* dependence was divided into three regions, within which the slope of the dependence is approximately constant: region (1) of n*-*type conductivity (*δ* < 0) and two regions of p*-*type conductivity 0 < *δ* < 0.005 (region 2) and 0.005 < *δ* < 0.011 (region 3). The obtained values of the hole mobility μ_p_ are 8.1 and 46 cm^2^ V^−1^ s^−1^ in regions 1 and 2, respectively. Thus, an increase in the silver content within the homogeneity region decreases hole’s mobility in the Ag_1.2±δ_Cu_0.8_Se. Similarly, for the region of electronic conduction, under the assumption that one free electron participating in the conduction corresponds to one excess silver ion Ag^+^ in the cation sublattice, the value of the electron mobility *μ**_n_* = 6.8 cm^2^ V^−1^ s^−1^ is obtained. For low temperature pseudo-tetragonal *β*-CuAgSe, in the paper of S. Ishiwata et al. [132], values of electron mobility were reported as ~20,000 cm^2^ V^−1^ s^−1^ at −263 °C and ~2500 cm^2^ V^−1^ s^−1^ at ambient temperature.

For hot-pressing consolidated CuAgSe nanoplatelets in work [133] by N.A. Moroz et al., it is observed that the thermopower of tetragonal *β*-CuAgSe gradually decreases with increasing temperatures from −50 µV K^−1^ at 27 °C to ~−20 µV K^−1^ at the vicinity of T_βα_ ~177 °C and jumps to a large positive value (+200 µV K^−1^) at a temperature slightly above *T_βα_* in the cubic *α*-CuAgSe. Earlier, Hong et al. [134] reported that high-temperature *α-*CuAgSe is a p-type semiconductor and exhibits low thermal conductivity, while *β-*CuAgSe shows metallic conduction with dominant n-type carriers and low electrical resistivity. The thermoelectric figure of merit *ZT* of the polycrystalline *α-*CuAgSe at 450 °C is ~0.95. The sign reversal from a negative value at low T to a positive value at high T was reported earlier [132], with the reversal temperature *T_S_* ~127 °C. This feature is not observed in Cu_2_Se and it reflects the existence of two types of carriers in the *β*-phase, as was claimed in work [132].

To determine the effective mass of electrons in the stoichiometric composition Ag_1.2_Cu_0.8_Se, the method developed by C. Wagner was applied in [130]. The value of the effective mass, at a temperature of 400 °C, found from processing the curves of coulometric titration according to the Wagner method [122], was *m** = 0.08 *m_e_*. This low effective mass is more characteristic for silver chalcogenides; much higher *m** values are observed for copper chalcogenides. Note that in Ag_2_Se, the effective mass of electrons was found equal to 0.10 *m_e_* in the state of equilibrium with selenium and 0.19 *m_e_* for the state of equilibrium with silver [135]. Thus, strong substitution with silver in copper selenide leads to the appearance of lighter charge carriers. Calculations in this work using the Mott formula show that the reduced Fermi level in the Li_0.10_Cu_1.75_Se alloy in the temperature range 20–60 °C has values *η* = 20 ÷ 23, and *η* = 24 ÷ 50 in the temperature range 130–230 °C.

Publications of experimental works on electron transport in compounds of the Na-Cu-S system are presented in small quantities, the first of which is the paper of Z. Peplinski et al. [136], published in 1982. It was obtained that from −258 °C to 27 °C, the mixed-valence compound Na_3_Cu_4_S_4_ is metallic. The compound exhibits Pauli paramagnetism with a value of *χ_m_* = 15 × 10^−5^ emu/mol in the temperature range of −173–27 °C. For pressed samples, the conductivity was measured from 300 S cm^−1^ at 27 °C to 1500 S cm^−1^ at −258 °C. Measurements on single crystals revealed that the conductivity σ is highly anisotropic, with enhanced conductivity *σ_‖_* parallel to the crystal needle axis, corresponding to pseudo-one-dimensional [Cu_4_S_4_^3-^]_∞_ columns in the structure. For single crystals, *σ_‖_* was measured from 15,000 S cm^−1^ at 27 °C to 300,000 S cm^−1^ at −258 °C. The authors of the work [136] assumed that the low values of conductivity observed for pressed tablets are the result of interparticle resistance. In the work of American scientists X. Zhang et al. [63] in 1995, the results of electrical conductivity for the composition NaCu_4_S_4_ are presented. The compound demonstrates metallic conductivity and temperature-independent Pauli paramagnetism with a value of *χ_m_* = 6.2 × 10^−5^ emu/mol. The conductivity decreases almost linearly with temperature, from ~3300 S/cm at −263 °C to ~240 S/cm at −48 °C, and the Seebeck coefficient is 3 μV/K, confirming the metallic nature of the p*-*type.

Interesting results of electrical and transport properties were obtained by Ge Z.H. [137] in 2016 for Na*_x_*Cu_1.8_S (*x* = 0, 0.005, 0.01, 0.03, 0.05). The obtained values of electrical conductivity (*σ*) of the samples are very high. The conductivity first increases and then decreases with temperature rise, where the turning moment is observed at ~100 °C. The *σ* values for Na_x_Cu_1.8_S samples continuously decrease with increasing Na content due to a decrease in the carrier concentration and mobility. The authors express the defect equation for Na entering the interstices of the Cu_1.8_S lattice as:(19)$$x\mathrm{Na}\overset{{\mathrm{Cu}}_{1.8}S}{\to }{\mathrm{Cu}}_{\mathrm{Cu}}^{x}+S_{s}^{x}+x{\mathrm{Na}}_{i}+xe^{\prime }$$

As shown in Equation (19), the Na^+^ ion enters the Cu_1.8_S lattice, increasing the electron concentration. The authors note that electrons recombine with holes, decreasing the hole concentration. The concentration of Hall carriers decreases from 6.37 × 10^22^ to 4.87 × 10^21^ cm^−3^ with *x* increasing. The mobility of carriers increases from 15.3 to 31.9 cm^−2^ s^−1^.

In the work of Zhu Z. et al. [53] in 2019, the electrophysical parameters are observed for Na*_x_*Cu_2-*x*_Se (*x* = 0, 0.01, 0.02, 0.03, and 0.04) bulk samples sintered by hot pressing, for both low- and high-temperature phases. The resistivity of Na*_x_*Cu_(2−*x*)_Se increases monotonically with increasing temperature, except for the inflection point near the phase transition temperature, which indicates that the sample exhibits metallic conductivity. It was shown that Na doping decreases the conductivity from 188 S cm^−1^ for Cu_2_Se to 130 S cm^−1^ for Na_0.04_Cu_1.96_Se at 700 °C. The authors [53] explain this behavior of the conductivity by a decrease in the carrier mobility caused by additional scattering from the numerous micropores of the structure. Together with this factor, undoubtedly, the reason for the decrease in the conductivity must be additional scattering of carriers by impurity sodium atoms.

We also note the results of the work of Zhang Yi et al. [138] published in 2020, devoted to the study of influence of Na_2_S doping (*x* = 0, 0.5, 1, 2 wt.%) on thermoelectric performance of digenite Cu_1.8_S in the temperature range 50–500 °C. It is well known that the pure composition of digenite demonstrates (Figure 3) extremely high electrical conductivity (*σ*) due to ionization of vacancies in the Cu sublattice. A transition from semiconducting to metallic conductivity is seen in Figure 4. The inflection point temperature is about 88 °C, which corresponds to the phase transition from the hexagonal phase to the cubic phase of digenite. The work was completed almost simultaneously with the work of Z. Zhu [53], i.e., independently, but what makes these works similar it is a high porosity of the material’s structure due to sodium doping. However, the material synthesized by Zhu turned out to be much more efficient (*ZT* ~1.3 at 500 °C) due to the fact that the thermal conductivity of his material was lower, while the factor *α^2^**σ* was almost close for both materials. It can be seen from the plots of the conductivity and the Seebeck effect in Ref. [138] that the concentration of charge carriers in the material is not optimized. It is possible that if not digenite, but jarleite (with the same 4 at.% Sodium) was chosen as the matrix for doping, the thermoelectric efficiency would be higher with the same low thermal conductivity due to the superionic state of the lattice and developed porosity.

Thus, in almost all works considered in this section, the substitution of lithium, sodium, and silver for copper reduces the electronic conductivity of copper chalcogenides due to a decrease in the carrier mobility and the effect of compensation of the hole concentration upon ionization of the donor impurity. In the case of heavy doping, the band gap and the effective mass of the carriers change. The doping of copper chalcogenides with an impurity of sodium sulfide revealed the formation of micropores, which weakly decreases the electrical conductivity and greatly decreases the thermal conductivity, generally improving the thermoelectric figure of merit *ZT*. Optimization of the carrier concentration by controlling the composition nonstoichiometry and impurity concentration remain the main methods for achieving the maximum thermoelectric power of copper chalcogenides and their alloys.

#### 3.1.2. Ionic Conductivity

Recently, solid-state ion-conducting superionic conductors have attracted increased attention of TE researchers. As is known, ionic transfer in ordinary solids does not exceed 10^−10^–10^−12^ S cm^−1^. However, an abnormally high ionic conductivity of superionic conductors ~1 S cm^−1^ is observed at temperatures significantly lower than their melting point, which is close to the conductivity of liquid electrolytes [19].

The crystal structure of the superionic phases of copper chalcogenides and similar compounds can be regarded as a rigid sublattice composed of chalcogen atoms (S, Se, Te) and disordered (melted) cationic sublattice, over which a liquid-like charged fluid of Cu ions diffuses [19,139]. In paper [1], the strategy was proposed to decrease lattice thermal conductivity below that of a glass by reducing not only the mean free path of lattice phonons but also eliminating some of the vibrational modes completely. According to authors [1], the idea of using the liquid-like behavior of superionic conductors may be considered an extension of the phonon glass electron crystal (PGEC) concept [140] and such materials could be considered phonon liquid electron crystal (PLEC) thermoelectrics. Local atomic jumps and rearrangement of the liquid inhibit the propagation of transverse waves and disrupt heat propagation by phonons [1]. Such liquid-like behavior results in ultralow lattice thermal conductivity and, for example, in the Cu_2_Se *α*-phase, it diminishes to 0.4–0.6 W m^−1^ K^−1^ and, as a result, provides the high *ZT*s around 1.3–2.1 at 727 °C [139].

Just as Drude’s idea of a gas of free electrons turned out to be fruitful for describing the current in metals, so the idea of “melting” of one of the crystal sublattices can become the foundation for the construction of a unified theory of diffusion in fast ionic conductors. The crowdion mechanism of diffusion has been known for a long time, but its implementation in ordinary crystals requires a very high activation energy of about 5–6 eV, which renders it negligible. However, something similar can occur in superionic conductors (SICs) under conditions of a “molten” sublattice of mobile ions. In a recent work by X. He et al. [141], it was shown by means of ab initio modelling for several lithium-ion conductors that “fast diffusion in superionic conductors does not occur through isolated ion hopping as is typical in solids, but instead proceeds through concerted migrations of multiple ions with low energy barriers”. To characterize the extent of concerted migrations, X. He et al. [141] calculated the correlation factor related to the Haven ratio. Whereas a correlation factor of 1.0 corresponds to isolated single-ion diffusion, the correlation factor is calculated as 3.0, 3.0, and 2.1 for Li_10_GeP_2_S_12_, Li_7_La_3_Zr_2_O_12_, and Li_1.3_A_l0.3_Ti_1.7_(PO_4_)_3_, respectively, in the AIMD simulations at 627 °C, corresponding to correlated hopping of approximately two to three ions on average in these SICs. Therefore, X. He et al. summarize that the concerted migration is the dominant mechanism for fast diffusion in SICs, as it is in liquids [142,143].

Since the crystal structure forms both the electronic and ionic properties of solids, they cannot be unrelated. Among other factors, lattice vibrations play an important role in the interaction of the electronic and ionic subsystems. Among the works on clarifying the relationship between the electronic, phonon, and ionic subsystems, in our opinion, the works of K. Wakamura [144] and H. Kikuchi et al. [145] should be noted. Wakamura [144] discusses the significant role of a narrow band gap in the formation of high ionic conductivity with low activation energy and low superionic transition temperature. He provides an analysis of experimental results for a wide class of materials, which shows the presence of a strong correlation between the above parameters. The high dielectric constant (for example, ε_∞_ = 9.7 for Cu_2_S at 23 °C) also correlates with high ionic conductivity, which Wakamura explains by the screening of the Coulomb interaction between ions by electrons. Kikuchi et al. [145] investigated the relationship between the ionic conductivity and the electronic structure in some copper and silver chalcogenides. It is known that Ag_2_Te has an antifluorite structure, and silver ions move along tetrahedral positions through adjacent octahedral positions. H. Kikuchi assumed that the antifluorite structure $Fm\bar{3}m$ is the ground state, and the local crystal structure $F\bar{4}3m$ can be considered as a transition state in the process of cation diffusion jumping. The diffusion jump changes the local crystal structure from *Fm3m* to $F\bar{4}3m$. Therefore, the activation energy of diffusion can be estimated as the difference between the binding energies of cations in the two aforementioned crystal lattices. It is noted that Ag_2_Te and Cu_2_Te have different degrees of *p-d* hybridization, and the *d* states of silver are much weaker associated with the *p* states of tellurium atoms and, therefore, are less localized. The random distribution of silver in the cation sublattice has no noticeable effect on the electronic structure of the ground state in Ag_2_Te. On the contrary, the band structure of Cu_2_Te is sensitive to the arrangement of copper atoms in the crystal lattice, which means a strong degree of *p-d* hybridization in Cu_2_Te. Thus, the ionic conductivity of Ag_2_Te higher than that of Cu_2_Te.

In the works of A.A. Lavrent’ev et al. [146] and Domashevskaya E.P. et al. [147], it was shown that, although the *d*-states of copper in chalcogenides are split in energy, and although they hybridize significantly with the *p*-states of anions, most of them fall into an energy region where there is a dip in the density of states of the anion and almost no hybridization. An increase in the symmetry of the environment of the anion leads to an increase in the fraction of copper *d* states not participating in hybridization. Copper selenide, which has a cubic anionic sublattice, has the lowest degree of hybridization. In our opinion, this means that copper is weakly bound to the anionic core and therefore has a low diffusion activation energy, as a result of which the high-temperature Cu_2_Se phase is an excellent superionic conductor for copper ions.

For investigations of superionic TE materials, electrochemical methods are very convenient. To measure partial electronic and ionic conductivities, to control and change the chemical composition of copper chalcogenides within their homogeneity region, the electrochemical cell contains electronic probes and electrodes (Pt) and ionic probes and electrodes (Cu/CuBr) [125,126] (Figure 4). This method is commonly referred to as the Hebb–Wagner method. By passing a direct current through circuit Cu/Cu_2−*δ*_X/CuBr/Pt, the copper deficit *δ* in Cu_2−*δ*_X (X = S, Se, Te) sample can be changed continuously with high precision (see above Equation (6)) and the e.m.f. *E* of the cell provides us with an information about the change in the chemical potential of copper atoms in the sample [122].

The voltage applied between the current electrodes at the conductivity measurements by Hebb–Wagner method must be lower than the decomposition potential of the phase under study (see Equation (16) above). An essential assumption underlying the applicability of the Hebb–Wagner method is the assumption that the chemical potential of copper ions in the sample becomes constant when a direct current is passed through the sample for a long time. To measure the electronic component of the conductivity of a mixed electronic–ionic conductor, it is necessary to measure the equilibrium value of the voltage drop across the electronic probes (Pt) when a direct current is passed through the electronic electrodes (Pt) [126]. When the current is switched on through the sample, the process of concentration polarization occurs in it, which is controlled by the chemical diffusion coefficient. For a 2 cm sample length, the equilibration time can range from a few seconds to several hours. If the ionic conductivity is comparable in magnitude to the electronic conductivity, then the error in the conductivity value using the nonequilibrium value of the potential difference between the potential probes can be large. This situation can be observed, for example, for the stoichiometric composition of copper sulfide in the superionic phase.

In order to search for ways to reduce the ionic conductivity, leading to the degradation of thermoelements, the substitution of copper in copper chalcogenides with lithium was investigated by Balapanov M. et al. [24], as well as the effect of grain sizes on the ionic conductivity [27,148]. It was found that in copper sulfide, the substitution of copper cations by lithium cations leads to a noticeable degradation in the properties of ion transport. In Table 1, parameters of ion transport in Li*_x_*Cu_2−*x*_S (0 ≤ *x* ≤ 0.25) superionic phases near a temperature of 355 °C are presented from [24].

Figure 5 shows the temperature dependences of the ionic conductivity for a few Li*_x_*Cu_2−*x*_S (*x* ≤ 0.25) solid solutions, from which it can be seen that the compositions Li_0.15_Cu_1.85_S and Li_0.2_Cu_1.8_S have the lowest ionic conductivity. If the ionic conductivity of Cu_2_S at 350 °C is 2.4 S cm^−1^ at an activation energy of 0.19 eV, then for Li_0.15_Cu_1.85_S, it is 0.26 S cm^−1^ at an activation energy of 0.45 eV. A significant decrease in ionic conductivity is a positive moment for the thermoelectric application of Li_0.15_Cu_1.85_S [149]. The authors [24] explain the concentration dependence of the ionic conductivity in the binary system Cu_2_S-Li_2_S with the well-known “mixed-mobile ionic effect”, which mainly is observed in superionic glasses, but it takes place in solids, too [150]. The chemical diffusion coefficients also tend to diminish with increasing lithium content due to a decrease in ionic conductivity. Ionic conductivity and chemical diffusion in the studied compounds have almost close values of the activation energy (Table 1). Substitution of lithium for copper in Cu_2-x_Li_x_S alloys led to a decrease in the temperature of the superionic phase transition and to an increase in the activation energy of ionic conductivity. The authors [24] suggest the absence of *d*-electrons in lithium as one of the reasons for the growth in the activation energy of the ionic conductivity. From an atomic crystal point of view, the reason for the deterioration of the diffusion properties of copper sulfide and copper selenide upon substitution with lithium is that lithium ions sadly occupy intermediate 32*f_II_* positions connecting tetrahedral and octahedral interstices in the anionic framework of the crystal lattice, through which channels of fast diffusion of cations pass. Overlapping of fast diffusion channels leads to an increase in the diffusion activation energy and a decrease in the cation mobility [26].

In works [151,152], it was found that copper and silver ions make comparable contributions to the total ionic conductivity of Cu_2_X-Ag_2_X solid solutions (where X = S, Se, Te). It was found that substitution by silver results in increasing ionic conductivity [151,152,153] of copper chalcogenides, while substitution by lithium leads to a strong decreasing in the ionic conductivity [22,24,26]. Balapanov M.Kh. et al. [154] studied the thermal diffusion of silver atoms and the Soret effect in Ag_(2−*x*)+*δ*_Cu*_x_*Se alloys (*x* = 0.1, 0.2, 0.4). The Soret effect (the gradient of metal concentration along the sample in a thermal field) was measured as *dδ/dT* = (0.2 ÷ 0.6) × 10^−5^ K^−1^. It was found that the heat of ionic transfer is much higher than the heat of electron transfer and is close to the activation energy of ionic conductivity.

In the work [27], the ionic conduction and the chemical diffusion in dependence on average grain sizes of Cu_1.75_Se, Li_0.25_Cu_1.75_Se, and Li_0.25_Cu_1.75_S samples at 140–240 °C were studied. The ionic conduction of Cu_1.75_Se is shown to increase with the average grain size. With the coarsening of grains, the activation energy for the ionic conduction decreases in Cu_1.75_Se and increases in Li_0.25_Cu_1.75_S. Usually for solids, grain boundary diffusion exhibits lower activation energy than bulk diffusion. The increase in the ionic conduction with the grain size in Cu_1.75_Se is explained by Balapanov M. [27] as the activation energy of cation diffusion through the grain bulk being lower than that for diffusion along the intergrain layers. Whereas in superionics with a high activation energy of the diffusion, grain boundaries provide accelerated diffusion, in the superionic conductor of the channel type with a low activation energy of diffusion throughout the grain bulk, the opposite occurs, since the fast diffusion channels are interrupted at the grain boundaries. Low activation energy here can be roughly considered *Е_а_* ~0.15 eV (*E_a_* = 0.14 eV for Cu_1.75_Se), for high conditionally, we can assume *Е_а_* ~0.40 eV and higher (*E_a_* = 0.39 eV for Li_0.25_Cu_1.75_S). *E_a_* value for the grain boundary diffusion in studied copper chalcogenides obviously lies between the above determined low and high activation energy of diffusion.

Thus, copper sulfide and selenide can be attributed to special superionic conductors, in which mobile ions move along the volume more easily than along grain boundaries. This leads to the conclusion that for thermoelectric applications at temperatures lower than ~200 °C, it is more advantageous to use copper sulfide and selenide with nanosized grains in order to reduce unwanted ionic conductivity.

Ionic conductivity and diffusion in sodium-doped copper chalcogenides has been poorly studied, if we do not take into account the works on the study of copper sulfide and selenide as active electrodes of sodium-ion batteries. In the work of M. Balapanov et al. [38], it is reported that the ionic conductivity of Na_0.2_Cu_1.8_S is about 2 S cm^−1^ at 400 °C (the activation energy *E_a_* ≈ 0.21 eV). The Seebeck ionic coefficient has high values of 0.3 ÷ 0.4 mV K^−1^. The thermal conductivity of superionic Na_0.2_Cu_1.8_S is low and provides high values of the dimensionless thermoelectric figure of merit *ZT* from 0.4 to 1 at temperatures from 150 °C to 340 °C.

High ionic and electronic conductivity of copper chalcogenides are very attractive to apply in chemical current sources. Sodium ion batteries (NIB) can become an alternative to lithium-ion batteries (LIB). Sodium is the fifth most abundant metal in the earth’s crust (2.27%). World prices for the main source of sodium—sodium carbonate—are ~20–30 times lower than the prices for lithium carbonate, the main raw material for the production of LIB components [155,156].

When the battery is operating, sodium ions are extracted from the material of the negative electrode and are embedded in the matrix of the positive electrode; during charging, the directions of the processes change. In works [157,158], the excellent properties of copper selenide Cu_2_Se as a cathode material for NIB are shown. The charge plateau is about 2.02 V during the charging process. The potential gap between the discharge and charge curves is only 0.1 V, which indicates very low polarization. The initial discharge capacity of the thin-film electrode is 253.0 mAh/g, and the charging capacity is 196.6 mAh/g. After 100 cycles, the discharge capacity was 113.6 mAh/g. The discharge power at 0.1 C, 0.5 C, and 2 C is 251.4, 122.8, and 90.8 mAh/g, respectively. These results indicate that Cu_2_Se is suitable for NIB use in fast charge/discharge mode. The electrochemical system sodium/copper sulfide (Na/Cu_2_S) was investigated by Kim J.-S. [159] using the 1M NaCF_3_SO_3_-TEGDME electrolyte. The first discharge curve of Na/Cu_2_S cells shows an oblique shape without a plateau potential region. The first discharge capacity is 294 mAh/g and decreases to 220 mAh/g after 20 cycles. The discharge process is explained by the intercalation of sodium into the Cu_2_S phase without separation of the Cu_2_S phases. Perhaps, high diffusion coefficients of copper sulfide and copper selenide will help to solve the serious problem of the sodium-ion electrochemical system, which is its very long charge/discharge time, because NIBs cannot yet operate at high current densities.

### 3.2. Seebeck Coefficient and Thermal Conductivity

#### 3.2.1. Seebeck Effect

In the Seebeck effect, the action of a temperature gradient in a material creates an electromotive force (e.m.f.). The ratio of this e.m.f. to the applied temperature difference is the Seebeck coefficient of the material. From a physical point of view, the Seebeck coefficient is the entropy transferred by a charge carrier during isothermal current flow, divided by the charge of the carrier [160]. Thus, the electronic Seebeck coefficient is directly related to the Fermi level of electrons. Usually, reduced Fermi level $\mu^{*}=\left(\mu -E_{c}\right)/k_{B}T$ and reduced energy of electrons $\varepsilon^{*}=\left(E-E_{c}\right)/k_{B}T$ are used in semiconductor physics, where *E_c_* is the carrier energy corresponding to bottom of conductivity band. For nondegenerate semiconductors ($\mu^{*}\ll 1)$, the Pisarenko formula for the Seebeck coefficient can be used [161]. In the case of carrier scattering by acoustic phonons in semiconductor with parabolic band, it looks as:(20)$$\alpha_{L}=\frac{k_{B}}{e}\left(2-\mu^{*}\right).$$

For carrier scattering by impurity ions, the Pisarenko formula is written as: (21)$$\alpha_{i}=\frac{k_{B}}{e}\left(4-\mu^{*}\right).$$

For degenerate semiconductors ($\mu^{*}>1)$ with a parabolic band, the Seebeck coefficient can be calculated as [161]:(22)$$\alpha_{L}=\frac{k_{B}}{e}\left(\frac{2F_{1}\left(\mu^{*}\right)}{F_{0}\left(\mu^{*}\right)}-\mu^{*}\right);\;\alpha_{i}=\frac{k_{B}}{e}\left(\frac{4F_{3}\left(\mu^{*}\right)}{3F_{2}\left(\mu^{*}\right)}-\mu^{*}\right).$$

In Equation (22), *F*_0_, *F*_1_, *F*_2_, and *F*_3_ are Fermi integrals, determined by general formula: (23)$$F_{n}=\overset{\infty }{\underset{0}{\int }}\frac{\varepsilon^{*}{}^{n}{d\varepsilon }^{*}}{1+\exp\left(\varepsilon^{*}-\mu^{*}\right)}.$$

For metallic systems ($\mu^{*}\gg 1)$ with a parabolic band, the Seebeck coefficient can be calculated as [161]:(24)$$\alpha_{L}=\frac{k_{B}}{e}\frac{\pi^{2}}{3}\frac{1}{\mu^{*}};\;\alpha_{i}=\frac{k_{B}}{e}\frac{\pi^{2}\mu^{*}}{\pi^{2}+\mu^{*}{}^{2}}.$$

The Seebeck coefficient in an impurity semiconductor with a parabolic band is proportional to the effective carrier mass and temperature and decreases with increasing carrier concentration (see Equation (15) above).

The Seebeck coefficient, thus, directly depends on the position of the Fermi level and on the concentration of charge carriers. The electron gas in metals is in a degenerate state; therefore, the Fermi level, the energy, and the electron velocity are weakly dependent on temperature. As a consequence, the values of the Seebeck coefficient are small. The Seebeck coefficient reaches relatively large values in semimetals and their alloys, where the carrier concentration is lower and depends on temperature. In this case, the concentration of electrons is high, nevertheless, the Seebeck coefficient is large due to the fact that the average energy of conduction electrons differs from the Fermi energy. Sometimes, fast electrons have a lower diffusion capacity than slow ones, and the Seebeck coefficient changes sign accordingly. The magnitude and sign of the Seebeck coefficient also depend on the shape of the Fermi surface. In some metals and alloys with a complex Fermi surface, various parts of the latter can give opposite sign to the thermo-e.m.f., and the Seebeck coefficient can be equal to or close to zero [162,163].

The Seebeck coefficient in semiconductors strongly depends on the scattering mechanism of charge carriers; it is seen from comparing Equations (21) and (22), for instance. In real crystals, several scattering mechanisms usually operate simultaneously. The contribution of each type of scattering can vary greatly depending on the temperature and concentration of impurities in the sample [162,163]. Accordingly, Formulas (21) and (22) heavy doping could lead to increasing the Seebeck coefficient at the same carrier’s concentration.

Binary copper and silver chalcogenides have excellent thermoelectric characteristics, which continue to be the subjects of intense research [2,4,82,83,109,139,164,165,166,167]. Over the past 5 years, as examples, we can mention the works [12,14,17,31,39,99,106,129,168,169] on Сu_2-__δ_S, the works [15,16,165,167,170,171,172,173,174,175,176,177,178] on Сu_2−*δ*_Se, the works [110,179,180,181,182,183,184,185] on Сu_2−*δ*_Te, the works [186,187,188,189,190] on Ag_2+*δ*_Se, the works [191,192,193,194] on Ag_2+*δ*_S, and the works [195,196,197,198,199] on Ag_2±*δ*_Te.

Some high achievements in thermoelectric performance of superionic chalcogenides and a number of other thermoelectric materials are shown in Figure 2 above and in Table 2 below.

##### Cu_2_Se

Resent achievements on Cu_2_Se-based thermoelectrics are carefully described in the review of Liu W.-D. et al., published in 2020 [165]. They state that very high *ZT* values close to 2 or higher have been achieved for Cu_2_Se thermoelectric materials within the temperature range from ~600 to ~700 °C [165]. The most significant achievement noted in their review is *ZT* = 2.6 at 577 °C for copper selenide doped with 1% CuInSe_2_ in 2017 [85]. In addition to describing thermoelectric performance, Liu W.-D. et al. summarize fundamentals of Cu_2_Se, including crystal structure, band structure, phase transition, and other aspects. Considering the abundance of Cu and Se and their high ZT values, Cu_2_Se-based thermoelectric materials are highly promising alternatives for the toxic PbTe thermoelectric materials, presume Liu W.-D. et al. [165]. 

It is well known that Cu_2_Se has an excellent intrinsic TE performance, however, practice shows that alloying with other elements can greatly improve the thermoelectric characteristics of copper selenide. Doping effect on Cu_2_Se thermoelectric performance was reviewed in 2020 by Y. Qin et al. [202]. The authors summed that through doping, many elements, such as Al, Li, Na, In, etc., have caused the *ZT* value of Cu_2_Se to exceed 2, and almost break through the minimum requirement of commercial use (*ZT* > 3). It is found that the high TE performance mainly originates from the fact that dopants can form point defects and dislocations, bring the mass fluctuation and strain fluctuation into the lattice, shift the microstructures, and so on, all of which play an important role in significantly scattering phonons and carriers (holes for Cu_2_Se, specifically), reducing the thermal conductivity below the glassy limit. Further, Y. Qin et al. suggest that, when doping Cu_2_Se, researchers can pay attention to the manipulation of microstructures, the anomaly at phase transition, and the avoidance of Cu deficiency, with which higher ZT values may be produced. In our opinion, the deficiency of copper in copper selenide should not be completely avoided, but it should be optimized, otherwise, the conductivity of the material may be insufficient to obtain the required thermoelectric power *α^2^**σ*.

In recent years, copper selenide has literally become a testing ground for many new ideas of researchers in thermoelectric materials. For example, D. Byeon et al. [175] in 2019 discovered the colossal Seebeck effect at the superionic phase transition in copper selenide, which was placed in a thermal field with two perpendicular temperature gradients (horizontal and vertical). The authors observed that Cu_2_Se shows two sign reversals and colossal values of *α* exceeding ±2 mV K^−1^ in a narrow temperature range, 67 °C < *t* < 127 °C, where a structure phase transition takes place. The metallic behavior of *σ* possessing larger magnitude exceeding 600 S cm^−1^ leads to a colossal value of power factor *α^2^σ* = 2.3 Wm^−1^ K^−2^. The small thermal conductivity less than 2 Wm^−1^ K^−1^ results in a huge dimensionless figure of merit exceeding 400.

For interpreting the unusual Seebeck coefficient, D. Byeon et al. [175] assumed that the chemical potentials of copper ions and conduction electrons in the low-temperature phase could be different from those in the high-temperature phase. In such a case, copper ions and electrons slightly move from one of the phases to the other so as to reach the energy equilibrium between both phases. The effect of the chemical potentials should change the carrier concentration of each phase. With an increasing temperature, the sample bottom starts transforming to the high-temperature phase, leading to an increase in the electron concentration in the low-temperature phase situated above due to the difference between the chemical potentials of electrons and/or copper ions. The number of electrons in the low-temperature phase in the top surface is further enlarged with the increased volume fraction of the high-temperature phase. Under the effect of the chemical potentials on the carrier concentration, the positive Seebeck coefficient of the low-temperature phase below 47 °C becomes negative at around 57 °C and reaches −4347 μV K^−1^ at 74 °C. In the higher temperature range above 74 °C, the chemical potential of either electrons or copper ions leads to a reduction in the electron concentration of the low-temperature phase, and the positive peak of the Seebeck coefficient of 1982 μV K^−1^ at 76 °C is created. In addition, one of the factors would be sustained in the temperature range of 77–97 °C, forming the plateau of the Seebeck coefficient (+220 μV K^−1^). The rather large value of *σ(T)* together with the large magnitude of the Seebeck coefficient naturally leads to surprisingly large values of power factor at the peaks: 0.18–2.3 W m^−1^ K^−2^ and 0.06–0.5 W m^−1^ K^−2^ for n-type and p-type, respectively. These values are definitely much larger than a few mW m^−1^ K^−2^ of typical thermoelectric materials.

The Equation (1) leading to the efficiency of energy conversion in a thermoelectric generator, *η*, that increases with an increasing *ZT*, was derived on the basis of a model involving *π*-type junctions comprised of two thermoelectric materials. In this scenario, a temperature gradient is applied to the thermoelectric materials simply along the direction of electrical current. The temperature distribution of the samples, in the setup introduced in work of D. Byeon et al. [175], was certainly different from the case of the *π*-type module. Therefore, *ZT* is no longer valid to estimate *η* in the case of two perpendicular temperature gradients, D. Byeon et al. [175] note. Nevertheless, it should be emphasized that the large *α(T)*, large *σ(T)*, and consequently obtained colossal *PF*, as measured using the same electrodes, can be certainly applicable. Of course, the narrowness of the temperature range in which the colossal Seebeck effect occurs and the condition of the presence of two cross-over temperature gradients will complicate the practical application of this interesting effect in thermoelectric devices. However, sometime technical problems will be solved, as, for example, it was with the use of liquid crystals in monitors, and, perhaps, a thermoelectric device based on this effect will be equally widespread.

##### Сu_2−*δ*_S

In 2014, He et al. [84] investigated the TE performances of Cu*_x_*S (*x* = 1.97, 1.98, and 2) and achieved very high *ZT*s of 1.4–1.7 at 727 °C. Similar with the *β*-Cu_2_Se phase with the character of “phonon-liquid electron-crystal”, [1] such high *ZT*s are mainly contributed by the ultralow lattice thermal conductivities (about 0.3–0.5 W m^−1^ K^−1^) caused by the liquid-like Cu ions. TE properties of copper sulfides with the Cu/S atomic ratios between 1.8 and 1.97 were studied in 2016 by P. Qiu et al. [203]. They reported results of investigations on crystal structures, valence states of elements, and thermoelectric properties of the compounds prepared by melting from elements at 1000 °C followed by hot pressing from fine powder. It was shown that the valence state of copper in these binary compounds does not change, and the thermoelectric properties were found to be very sensitive to copper deficiency. In addition, they reveal that the arrangement of sulfur in the crystal structure also plays an important role in electrical transfer. The optimal compositions of Cu_2-x_S were determined to obtain a high power factor and thermoelectric figure of merit. The crystal structure of the investigated compositions, namely the valence states of copper and sulfur in all compositions, had similar values. At 473 °C, the maximum power factor was 12.5 μW cm^−1^ K^−2^ in Cu_2−*x*_S (2–*x* = 1.90–1.92), and the maximum *ZT* = 0.8 was observed for the Cu_1.96_S composition with total thermal conductivity *κ* ≈ 1 W m^−1^ K^−1^. Due to the lower κ, Cu_1.97_S and Cu_1.98_S reported by He et al. [84] possess higher *ZT* than Cu_1.96_S at 477 °C in [203].

Taking into account its own ultra-low lattice thermal conductivity, an increase in the performance of *TE* in Cu_2_S can be achieved by improving its electrical transport properties. Nanosized forms of copper sulfides (platelets, discs, rods, and others) promote their use at a more advanced level, adjusting their properties depending on the shape and size of the particles of materials. Large-scale Cu_2_S tetradecahedrons microcrystals and sheet-like Cu_2_S nanocrystals were synthesized by employing a hydrothermal synthesis (HS) method and wet chemistry method (WCM), respectively, by Yun-Qiao Tang et al. [14] in 2017. The polycrystalline copper sulfides bulk materials were obtained by densifying the as-prepared powders using the spark plasma sintering (SPS) technique. The pure Cu_2_S bulk samples sintered by using the powders prepared via HS reached the significant thermoelectric figure of merit (*ZT*) value of 0.38 at 300 °C. Contrary to hydrothermal sintered Cu_2_S bulk samples, the highly dense Cu_1.97_S bulks, fabricated by a melt-solidification technique by L. Zhao et al. [17], showed higher thermoelectric performance with *ZT* of 1.9 at 697 °C.

Often, introducing suitable impurities helps to enhance useful properties of the TE material. Guan M. et al. [129] carried out doping of Cu_2_S with lithium. Series of Cu_2−*x*_Li*_x_*S samples with various Li contents (*х* = 0, 0.005, 0.010, 0.050, and 0.100) were synthesized by fusion with subsequent annealing. At the composition of *x* = 0.05, the sample was more stable, and homogeneous phases with the same monoclinic structure as Cu_2_S at room temperature were found. It was revealed that the electrical conductivity in the Cu_2−*x*_Li*_x_*S samples is significantly improved by the introduction of Li; the electrical conductivity increases due to the increased concentration of carriers. The maximal figure of merit *ZT* = 0.84 was achieved for Cu_1.99_Li_0.01_S composition at 627 °C, about a 133% improvement as compared with that in Cu_2_S matrix.

Work on doping copper sulfide and copper selenide with lithium should be continued with the choice of other compositions of the initial chalcogenide, since the possibilities of increasing the thermoelectric power of the material seem to be higher, in our opinion, than was achieved in the described works. This was also shown in 2018 by the subsequent work of Hu et al. [200], in which for the Li_0.02_Cu_1.98_S composition, the value *ZT* = 2.14 was achieved, which exceeds the indices of pure copper sulfide.

The use of the electrochemical method of doping with lithium made it possible to obtain and study solid solutions Li*_x_*Cu_2−*x*_S (0 < *x* < 0.25) and Li*_x_*Cu _(2−*x*)−*δ*_Se (*x* ≤ 0.25), which are promising for thermoelectric applications [23,24]. The authors of [149] investigated a semiconductor alloy with the composition Li_0.15_Cu_1.85_S with an ionic conductivity 8–10 times lower than that of pure copper sulfide, characterized in that it was obtained by cold pressing from Li_0.15_Cu_1.85_S nanopowder. At room temperature, the alloy is heterophase, consisting of an orthorhombic Cu_1.75_S phase, a tetragonal Cu_1.96_S phase, a hexagonal Cu_2_S phase, and a cubic Cu_2_S phase.

##### Сu_2-δ_Te

Pure copper telluride demonstrates more modest thermoelectric characteristics compared to copper selenide and sulfide. It may be caused the fact that after the same SPS sintering, copper telluride Cu_2−*δ*_Te usually has higher electrical conductivity and lower thermopower in comparison with Cu_2−**δ**_Se or Cu_2−*δ*_S, owing to its greater copper deficiency [110].

He Y. at al. [110] in 2015 investigated high-density Cu_2_Te samples obtained using direct annealing without a sintering process (SPS). In the absence of sintering processes, the samples’ compositions could be well controlled, leading to substantially reduced carrier concentrations that are close to the optimal value. The electrical transports were optimized, and the best power factor was achieved near 1300 μW m^−1^ K^−2^, which is more than a 30% enhancement when compared with the sample sintered using SPS. The minimal thermal conductivity at 327 °C was reduced to ~0.7 W m^−1^ K^−1^ for direct annealing sample instead of ~1.2 W m^−1^ K^−1^ for SPS sintered sample. The *ZT* values were significantly improved by He Y. at al. [110] to 1.1 at 727 °C, which is nearly a 100% improvement compared with SPS sintered sample. Furthermore, He Y. at al. note that this method saves substantial time and cost during the sample’s growth.

Cu_2_Te/Te nanorod composites were fabricated by ultrasonication in ethanol from a mixture of Te and Cu_2_Te nanorods and their thermoelectric properties were investigated by D. Park et al. [181]. The thermoelectric power factor of the Cu_2_Te/Te nanorod composites at room temperature was the highest (431 μV K^−1^) at 10 wt% Te, and the enhancement in power factor is achieved to ~440 μW m^−1^ K^−2^ for this composition. The authors note that a great reduction in the total thermal conductivity *κ* was caused by the strong phonon scattering effect owing to a 1D nanostructure of the composites. Because of the enhancement in power factor and decrease in thermal conductivity, the composite samples showed an enhanced thermoelectric figure of merit *ZT* = 0.22 at room temperature, which was achieved for a Te content of 10 wt% and was ~4.5 times larger than that of the pristine Cu_2_Te nanorods. D. Park et al. highlight the fact that the composites with the two types of nanorods form Cu_2_Te/Te homo-interfaces and show a lower *κ* value than the two types of pristine nanorods, because they form an effective phonon-scattering center. This is one of many results that support our thesis about the desirability of greater attention of researchers to composite thermoelectric materials, including due to possible positive synergetic effects.

Mukherjee S. et al. [182] studied thermoelectric properties of Cu_2-*x*_Fe*_x_*Te solid solutions until the solubility limit of Fe ~*x* = 0.05 and obtained a maximum figure of merit *ZT* ~0.16 at 477 °C for Cu_1.97_Fe_0.03_Te composition. The authors revealed that the specific heat capacity *C_p_* of Fe alloyed samples higher than the Dulong–Petit limit of 3 *Nk_B_*, and the minimum thermal conductivity (~2.19 W m^−1^ K^−1^) is high for superionic conductor. Mukherjee S. et al. [182] supposed that the superionic “liquid-like” behavior of the Cu ions is suppressed by the presence of Fe, which is rather probable. On the contrary, doping with a more pronounced superionic nanostructured Ag_2_Te conductor made it possible for Mukherjee S. et al., in their next work [183], to sharply reduce the thermal conductivity of bulk copper telluride from ~3.5 *W* m^−1^ K^−1^ to ~0.26 W m^−1^ K^−1^ and achieve the high *ZT* ~0.99 at 315 °C for (Cu_2_Te)_50.00_-(Ag_2_Te)_50.00_ composition. The authors explain the achievement as due to the additional scattering of carriers as well as phonons by the Ag_2_Te nanostructures. It can be clarified here that, possibly, the thermal conductivity strongly decreases both due to the liquid-like state of the lattice, since silver telluride is a fast ionic conductor (the ionic conductivity ~1.6 S cm^−1^ at 300 °C, the activation energy of the ionic conductivity E_a_ = 0.13 eV) [184], and due to additional scattering of phonons at interphase boundaries, the area of which greatly increases in the presence of silver telluride nanoparticles.

After long attempts, *ZT* was also brought to the level of 1.5 for copper tellurides. This was achieved by the group of Zhao K. et al. [185]. Doping with silver telluride also brought success. It is demonstrated that the Cu_2_Te-based compounds are also excellent TE materials if Cu deficiency is sufficiently suppressed. By introducing Ag_2_Te into Cu_2_Te, the carrier concentration is substantially reduced to significantly improve the *ZT* with a record-high value of 1.8, a 323% improvement over Cu_2_Te, and outperforming any other Cu_2_Te-based materials. The single parabolic band model is used in [185] to prove that all Cu_2_X-based compounds are excellent TE materials.

The highest *ZT* reported by Zhao et al. [185] for Cu_2_Te 50% Ag_2_Te is ~1.8 at 727 °C. Similarly, inclusion of nanosized Ag_2_Se into Cu_2_Se increased the *ZT* to ~0.9 at 315 °C and to ~1.85 at 527 °C, according to report of Ballikaya et al. [176]. In the work of Mukherjee S. [183], the incorporation of nanostructured Ag_2_Te into bulk Cu_2_Te obtained by the melting route also improved the thermoelectric properties of Cu_2_Te significantly. The value *ZT* = 0.99 at 315 °C obtained in [183] for (Cu_2_Te)_100−*x*_-(Ag_2_Te)*_x_* composites (where *x* varies from 0 to 50) is sufficiently higher than that reported for the alloy Cu_2_Te 50% Ag_2_Te by Zhao et al. (*ZT* ~0.79) at the same temperature [185]. Thus, the mechanical alloying of Cu_2_Te with nanostructured Ag_2_Te, as followed in the [183], is an effective method in enhancing the TE performance of Cu_2_Te compared to the conventional vacuum melting route. Given the results of Zhao et al. [185], Ballikaya et al. [176], and Mukherjee S. [183], one can conclude that Cu_2_X-Ag_2_X (X = S, Se, and Te) composites are promising thermoelectric materials for mid-temperature applications.

Sodium doping in copper chalcogenides was studied in works [31,36,37,38,39,40,53,138]. Copper chalcogenides with a high sodium content are not homogeneous and form a mixture of phases with different conducting and thermoelectric properties, but deserve attention as nanocomposite thermoelectric materials with low thermal conductivity. The properties of such mixtures are poorly understood; however, we expect new effects here associated with the presence of numerous barrier layers and pores.

In alloys of the Na-Cu-S system, the results show that Na plays two important roles: the first reduces the concentration of carriers, thereby improving the Seebeck coefficient, the other reduces the thermal conductivity, which is generally favorable, improving the characteristics of the thermoelectric figure of merit [62,138].

Potassium doping in copper selenide was studied by Z. Zhu et al. [171] using hydrothermal synthesis and hot pressing. Numerous micro-pores were introduced by K doping, together with reduced electronic conductivity that result in low thermal conductivity. For the nominal component Cu_1.97_K_0.03_Se (EPMA measured composition Cu_1.99_K_0.01_Se), the peak value of *ZT* reaches 1.19 at 500 °C, which is 47% larger than that of pure Cu_2_Se (*ZT_max_* = 0.81).

For comparison, we present several works on other groups of thermoelectric materials. Bismuth telluride-based solid solutions and bismuth telluride nanocrystals also remain in field of attention [92,98].

Half-Heusler alloys are intensively studied to improve thermoelectric performance [111,112,113]. Half-Heusler phases (space group $\bar{4}3m$, C1b) have recently captured much attention as promising thermoelectric materials for heat-to-electric power conversion in the mid-to-high temperature range. The most studied ones are the RNiSn-type half-Heusler compounds, where R represents refractory metals Hf, Zr, and Ti. These compounds have shown a high-power factor and high-power density, as well as good material stability and scalability. Due to their high thermal conductivity, however, the dimensionless figure of merit (*ZT*) of these materials has stagnated near 1 for a long time. Since 2013, the verifiable *ZT* of half-Heusler compounds has risen from 1 to near 1.5 for both n- and p-type compounds in the temperature range of 500–900 °C [111].

Skutterudites represent a promising group of thermoelectric materials [83,114,116,117]. Skutterudites are bulk TE materials with the crystal structure belonging to cubic space group *Im3*. They contain vacancies into which low-coordination ions (usually rare earth elements) can be inserted to decrease the thermal conductivity by enhancing phonon scattering without reducing electrical conductivity. Such a structure makes them exhibit PGEC (phonon glass electron crystal) behavior. *ZT* of skutterudites can be significantly improved by double, triple, and multiple filling of elements into the vacancies in their structure. Usually, alkali metals, alkaline earth metals, lanthanides, and similar elements are selected as “fillers” or “dopants” for skutterudites because of their moderate atom size. Hence, the selection and combination of different “fillers” is very crucial for achieving high TE performance. Replacement of host atoms in lattice sites, such as substitution for Co and Fe, has also been identified as an effective way to reduce thermal conductivity [164].

J. Chen et al. [190] in 2020 wrote, owing to the intrinsically good near-room-temperature thermoelectric performance, orthorhombic *β*-Ag_2_Se has been considered as a promising alternative to n-type Bi_2_Te_3_ thermoelectric materials. Herein, we develop an energy- and time-efficient wet mechanical alloying and spark plasma sintering method to prepare porous *β*-Ag_2_Se with hierarchical structures including high-density pores, a metastable phase, nanosized grains, semi-coherent grain boundaries, high-density dislocations, and localized strains, leading to an ultralow lattice thermal conductivity of ~0.35 W m^−1^ K^−1^ at 27 °C. Relatively high carrier mobility is obtained by adjusting the sintering temperature to obtain pores with an average size of ~260 nm, therefore resulting in a figure of merit, *ZT*, of ~0.7 at 27 °C and ~0.9 at 117 °C. The single parabolic band model predicts that *ZT* of such porous *β*-Ag_2_Se can reach ~1.1 at 27 °C if the carrier concentration can be tuned to ~1 × 10^18^ cm^−3^, suggesting that *β*-Ag_2_Se can be a competitive candidate for room-temperature thermoelectric applications.

In the end of this section, let us mention the unusual thermoelectric studies that were published in recent years. G. Kim et al. reported in [196] about a new interesting method of control of the thermoelectric properties of polycrystalline Ag_2_S by spatial-phase separation into low- and high-temperature phases using a configuration based on bottom heating and top measurement. The authors [196] experimentally confirmed that the Seebeck coefficient was determined by the low-temperature phase at the top surface but the electrical resistivity was dominated by the high-temperature phase lying below the low-temperature phase. As a result, a high Seebeck coefficient ~−0.650 mV K^−1^ and high conductivity ~500 S cm^−1^ were simultaneously observed over a broad temperature range (17–167 °C). The authors [196] hope that this idea suggests a new concept for thermoelectric materials and devices. Of course, the technical implementation of such a thermoelectric device is a big problem, but the observed effect is impressive and deserves attention. There is some similarity of this new approach to increasing the thermoelectric efficiency of materials with another original work by Japanese researchers [175], in which the use of a second transverse temperature gradient allows one to obtain a colossal thermoelectric effect near the temperature of the superionic phase transition in copper selenide.

In addition to ordinary rigid *TE*, the flexible thermoelectric materials are objects of greatly increasing attention in the past decade [167], and a variety of conductive polymers, carbon materials, nano-sized inorganic semiconductors, and metals have been applied to fabricate flexible thermoelectric films. So far, in spite of their relatively poor flexibility, the nano-sized inorganic semiconductors are superior to their counterparts because of their excellent thermo-electric performance [186]. For example, silver selenide is considered as a promising room-temperature thermoelectric material due to its excellent performance and high abundance. However, the silver selenide-based flexible film is still behind in thermoelectric performance compared with its bulk counterpart. In work by J. Gao et al. [186], the composition of paper-supported silver selenide film was successfully modulated through changing reactant ratio and annealing treatment. In consequence, the power factor value of 2450.9 ± 364.4 mW/(mK^2^) at 30 °C, which is close to that of state-of-the-art bulk Ag_2_Se, has been achieved. On base of the film, the thermoelectric device was established. At a temperature difference of 25 °C, the maximum power density of this device reaches 5.80 W/m^2^, which is superior to that of previous film thermoelectric devices. Similar investigations of silver selenide were performed by Ding Y. et al. [187] and Perez-Taborda J.A. et al. [189]. Recently, Liang J. et al. [204] reported high intrinsic flexibility and state-of-the-art figures of merit (up to 0.44 at 27 °C and 0.63 at 177 °C) in Ag_2_S-based inorganic materials, opening a new avenue of flexible thermoelectrics. In the flexible full-inorganic devices composed of such Ag_2_S-based materials, high electrical mobility yielded a normalized maximum power density up to 0.08 W m^−1^ under a temperature difference of 20 °C near room temperature, orders of magnitude higher than organic devices and organic–inorganic hybrid devices.

Theoretically and experimentally, these works exerted a significant effort to achieve silver chalcogenide-based flexible thermoelectric films and devices with high performance to provide electricity to electronic devices for an individual’s timely healthcare, real-time safety monitoring, and life improvement by utilizing the temperature difference between the skin and the ambient environment, all-weather, regardless of the motion state of the human body.

#### 3.2.2. Thermal Conductivity

The corpuscular or phonon approach to the consideration of lattice vibrations is especially convenient in the study of energy conversion processes. These processes include the processes of phonon creation and annihilation. It is most convenient to describe thermal conductivity in terms of phonon scattering by other phonons, static imperfections of the lattice, or by electrons [124]. The electronic structure and dynamics of the lattice are of particular importance for thermoelectric materials. The work of Bikkulova N.N. [205] first studied the lattice dynamics in the superionic and nonsuperionic states for solid solutions of copper chalcogenides. It was found that in superionic conductors, the main vibration modes are of an acoustic nature. Resonant interaction of the *d-p* state of anions in the valence band leads to screening of the electronic field of cations, lowers the activation barrier, and promotes disordering of the cation sublattice. Considering that the thermal conductivity of superionic copper chalcogenides is low and does not depend too much on the composition, it can be predicted that at temperatures 30–127 °C, materials based on lightly doped copper will be the most effective in thermoelectric devices. Nanostructuring of a material can further reduce thermal conductivity due to phonon scattering on grain boundary inhomogeneities and increase thermoelectric efficiency.

The enhancement of the effect of increasing the thermoelectric figure of merit in copper chalcogenides is achieved with nanostructuring, and the thermal conductivity decreases significantly more than the conductivity. This is due to the fact that in the formation of the optimal structure of the material, it is necessary to create conditions for phonons to be scattered more strongly by structure inhomogeneities than electrons. Since the wavelengths of electrons and phonons are different, one of the factors that can be used here is the size factor, since scattering is enhanced when the de Broglie wavelength becomes comparable to the size of the inhomogeneities in the medium.

Debye introduced the concept of the mean free path *l* and obtained a formula for a thermal conductivity similar to the formula following from the kinetic theory of gases:(25)$$k_{L}=\frac{1}{3}c_{V}\bar{v\bar{l}}$$
where $c_{V}$ is the heat capacity of 1 *см^3^* of the crystal, $\bar{v}$ is the average sound speed in the crystal, and $\bar{l}$ is the phonon mean free path.

Peierls changed Debye’s theory, showing that the establishment of thermal equilibrium in a system of phonons can be influenced by processes for which the energy conservation law is satisfied:(26)$$ћ\omega_{1}+ћ\omega_{2}=ћ\omega_{3}$$
(27)$$ћk_{1}+ћk_{2}=ћk_{3}$$

The wave vectors are related by:(28)$$k_{1}+k_{2}=k_{3}+G$$

These are the laws of conservation of energy and momentum (N-processes). Here, *G* is the reciprocal lattice vector. Since a phonon with a wave vector (+*G*) in a periodic lattice is indistinguishable from a phonon with a wave vector, Peierls called such processes Umklapp or U-processes. A feature of U-processes is that they change the momentum and direction of the energy flow. Thus, the U and N processes are responsible for the formation of thermal resistance to the phonon flux and ensure the establishment of thermal equilibrium in the phonon distribution [201].

For a known Fermi level and carrier scattering mechanism, the electronic component of thermal conductivity *k_e_* can be calculated through the Fermi integrals using the formulas [206,207]
(29)$$K_{e_{a}}={\left(\frac{k_{B}}{e}\right)}^{2}\left(\frac{3F_{0}\left(\mu^{*}\right)F_{2}\left(\mu^{*}\right)-4F_{1}{}^{2}\left(\mu^{*}\right)}{F_{0}{}^{2}\left(\mu^{*}\right)}\right);\;K_{e_{i}}={\left(\frac{k_{B}}{e}\right)}^{2}\left(\frac{15F_{2}\left(\mu^{*}\right)F_{4}\left(\mu^{*}\right)-16F_{3}{}^{2}\left(\mu^{*}\right)}{9F_{2}{}^{2}\left(\mu^{*}\right)}\right).$$
where $\mu^{*}$ is the reduced Fermi level, $K_{e_{a}}$ and $K_{e_{i}}$ are electronic thermal conductivities for scattering by acoustic phonons and by impurity ions, respectively, and the Fermi integrals $F_{n}\left(\mu^{*}\right)$ are given by Equation (23) above. To determine the Fermi level, one usually uses the experimental values of Seebeck coefficient and theoretical expressions for the Seebeck coefficient (see Equation (22)).

As already noted, the interaction of the electronic, ionic, and phonon subsystems of the crystal is well manifested in copper chalcogenides [20,144]. Thus, the “liquid-like” cationic sublattice is responsible for very low thermal conductivity and nonstoichiometric defects determine high electronic conductivity, which contributes to the high thermoelectric figure of merit of copper chalcogenides [1,144,173].

Lattice vibrations in crystalline materials generate phonons as heat carriers for heat conduction, and the phonon dispersion (energy versus momentum) is fundamentally determined by the mass of lattice vibrators (atoms) and the interaction force between atoms. A significant manipulation of lattice thermal conductivity through a change in atomic mass usually requires a large variation in chemical composition, which is not always valid thermodynamically or may risk the resultant detriment of other functionalities (e.g., carrier mobility). Here, Wu et al. [208] show a strategy of alternatively manipulating the interaction force between atoms through lattice strains without changing the composition, for remarkably reducing the lattice thermal conductivity without reducing carrier mobility, in Na_0.03_Eu_0.03_Sn_0.02_Pb_0.92_Te with stable lattice dislocations. This successfully leads to an extraordinarily high thermoelectric figure of merit *ZT* = 2.4 at 527 °C, with the help of valence band convergence. This work offers both insights and solutions on lattice strain engineering for reducing lattice thermal conductivity, thus advancing thermoelectrics.

In [209], Liu et al. reported results showing that binary ordered Cu_2-δ_Se has an extremely low lattice thermal conductivity at low temperatures. The low energy multi-Einstein optic modes are the dominant approach to obtaining such an extremely low lattice thermal conductivity. It is indicated that the damped vibrations of copper ions could contribute to the low energy multi-Einstein optic modes, especially for those low energy branches at 2–4 meV. Recent work on Cu_2−*δ*_X (X = S, Se, or Te) thermoelectric materials [15] reveals the ultralow lattice thermal conductivity (0.3–0.6 W m^−1^ K^−1^) with the reduced specific heat down to the limit of a solid material at high temperatures. The “liquid-like” diffusion of disordered copper ions is believed to be the origin of the abnormal and interesting thermal transport phenomenon. In addition, these materials also exhibit extremely low thermal conductivity at low temperatures at which the copper ions display highly ordered distributions. The ultralow lattice thermal conductivity in the well-ordered simple binary Cu_2−*δ*_Se compounds below room temperature is quite abnormal and special, but the mechanism is still unknown. The lattice dynamics of several copper based thermoelectric materials were performed by studying the heat capacity and using inelastic neutron scattering techniques [210,211], demonstrating that those low energy localized vibrational modes mainly form the motion of copper atoms.

In 2021, Dutta M. et al. [212] presented lattice dynamics associated with the local chemical bonding hierarchy in Zintl compound TlInTe_2_, which cause intriguing phonon excitations and strongly suppress the lattice thermal conductivity to an ultralow value (0.46–0.31 W m^−1^ K^−1^) in the range 30–400 °C. They established an intrinsic rattling nature in TlInTe_2_ by studying the local structure and phonon vibrations using synchrotron X-ray pair distribution function (PDF) (−173–230 °C) and inelastic neutron scattering (INS) (−268–177 °C), respectively. They showed that during 1D chain of covalently bonded ${\left[InTe_{2}\right]}_{n}^{-n}$ transport heat with Debye type phonon excitation, ionically bonded Tl rattles with frequency ~30 cm^−1^ inside a distorted Thompson cage formed by ${\left[InTe_{2}\right]}_{n}^{-n}$. This highly anharmonic Tl rattling causes strong phonon scattering and consequently, phonon lifetime reduces to an ultralow value of ~0.66(6) *ps*, resulting in ultralow thermal conductivity in TlInTe_2._ The temperature-dependent X-ray PDF and INS investigations provided conclusive evidence the origin of low thermal conductivity in Zintl TlInTe_2_. Thus, critical examination of chemical bonding, local structure, and experimental determination of phonon DOS should be the way forward to explore the fundamental origin of low thermal conductivity in crystalline material.

In the recent works of Bulat et al. [171,172], it was found that the experimental values of thermal conductivity in nanostructured samples of copper selenide are significantly lower than those given by theoretical calculations. According to the results of electron microscopic (high resolution) studies [172], a large number of nanosized grains and defects are observed in nanostructured copper selenide. This allowed the authors to conclude that the extremely low lattice thermal conductivity is due to phonon scattering at these intergrain nanoboundaries and nanodefects. This conclusion is confirmed by the results of studies where in nanostructured samples synthesized by a chemical method followed by SPS, the lattice thermal conductivity was 0.2 [173] and 0.23 W m^−1^ K^−1^ [174] at 577−627 °C. The presence of such nanodefects was not taken into account in the calculations of thermal conductivity. This is precisely the reason for the significant difference between the calculated and experimental values of the lattice thermal conductivity at temperatures above 507 °C.

In [213], Matthias Agne et al. expressed his concerns about the use of the well-known formula for calculating thermal conductivity. The ease and access of thermal diffusivity D measurements allows for the calculation of *κ* when the volumetric heat capacity, *ρc_p_*, of the material is known. However, in the relation *κ = ρc_p_D*, there is some confusion as to what value of *c_p_* should be used in materials undergoing phase transformations. In this paper, it is demonstrated that the Dulong–Petit estimate of *c_p_* at high temperature is not appropriate for materials having phase transformations with kinetic timescales relevant to thermal transport. In these materials, there is an additional capacity to store heat in the material through the enthalpy of transformation *ΔH*. This can be described using a generalized model for the total heat capacity for a material *ρc_p_ = C_ρφ_ + ∆H**∙(∂φ/∂T)_p_*, where *φ* is an order parameter that describes how much latent heat responds “instantly” to temperature changes. Here, *C_pφ_* is the intrinsic heat capacity (e.g., approximately the Dulong–Petit heat capacity at high temperature). It is shown experimentally in Zn_4_Sb_3_ that the decrease in D through the phase transition at −23 °C is fully accounted for by the increase in *c_p_*, while *κ* changes smoothly through the phase transition. Consequently, conclude Agne et al. [213], reports of *κ* dropping near phase transitions in widely studied materials such as PbTe and SnSe have likely overlooked the effects of excess heat capacity and overestimated the thermoelectric efficiency, *ZT*.

An incomplete understanding of heat capacity measurements and models can lead to inaccurate estimations of *κ* in some systems, especially those having substantial latent heats (e.g., during phase transitions). The recent debate surrounding the thermoelectric material Cu_2_Se is an excellent example [214,215,216,217,218]. In this material and others [91,219], the thermal diffusivity drops markedly as the material undergoes a phase transition. Depending on the heat capacity used to calculate *κ*, a maximum *ZT* between 0.6 [220] and 2.3 [214] has been reported due to the superionic phase transition in Cu_2_Se. It is exemplified in the paper [213] that the choice of heat capacity can have a drastic impact on ZT values. Recognizing that the total capacity of a material to absorb heat includes both the intrinsic heat capacity of the phases present and the enthalpy (heat) of transformation *ΔH* that is required to maintain equilibrium (characterized by the order parameter *φ*), the temperature is changed as: (30)$$\rho c_{p}={\left(\frac{\partial H}{\partial T}\right)}_{p}=C_{\rho \phi }+\Delta H\cdot {\left(\frac{\partial \phi }{\partial T}\right)}_{p}.$$

This is understandable, as one could argue that *(∂φ/∂T)_p_* should be zero in the steady-state measurement of *κ*. However, it is shown theoretically in the article by Agne et al. [214] why this term is non-zero and should be included in the calculation of *κ* when transformation kinetics are fast on the timescale of thermal transport. Not including the enthalpy of transformation can lead to significantly underestimated values of *κ* in the region of peak *ZT* for many important cases, such as Cu_2_Se, PbTe, and SnSe, where exceptional *ZT* > 2 has been reported. In many good thermoelectric materials at their operation temperatures, atomic rearrangement may be fast enough that latent heats will suppress the thermal diffusivity. An apparent increase in *ZT* will result if a heat capacity is used that does not account for the latent heats. In particular, any discontinuity, spike, or sharp decrease that is found in thermal diffusivity measurements should be scrutinized before the same features are ascribed to the thermal conductivity. Even estimated values of lattice thermal conductivity *κ_L_* which are substantially below estimates for the lower limit of thermal conductivity [221] should be scrutinized as they may indicate that *κ* is underestimated, as was found in cases of dynamic doping. Substantial underestimates of *κ* and overestimates of *ZT*, as demonstrated in Cu_2_Se, are likely prevalent in other systems such as SnSe [213].

The crystal structure and various physical properties of semiconductor compounds of copper sulfides have been studied for a long period of time, starting from the 1930s; therefore, there is a need for new research in this area using modern research methods, since equipment and methods have emerged that make it possible to take a new look at the already established facts and discover new sides of known structures and phenomena. Differential thermal analysis (DTA) and differential scanning calorimetry (DSC) are among the most sensitive and reliable methods for studying the thermal properties of solids, which are often used in practice, as they allow one to determine all the main properties of heat transfer, specific heat, enthalpy, and phase equilibria of structural transition.

The thermal properties of copper sulfides have been less studied than other physical properties. The earliest studies of the thermal and thermoelectric properties of Cu_2-x_S were carried out on polycrystalline and single-crystal samples [47,222]. It was shown that the properties of Cu_2-x_S strongly depend on the nonstoichiometry *x* of the composition. In addition, the thermal properties were found to exhibit anomalous behavior near the temperatures of the phase transformation in Cu_2-x_S. The results show that the mechanism of heat transfer is mainly due to phonons, while the contribution of electrons and dipoles was indeed very small [222].

The heat capacity and thermodynamic properties of Cu_2_S in the temperature range 268–677 °C were studied in detail by F. Gronvold [223]. The phase transitions were recorded at about 103 and 437 °C. Numerical results on the enthalpies in the transitions *(*γ → β) and (β → α*)* in Cu_2_S are given [223]. Significantly higher values of heat capacity *C_p_* ≈ 11.9 *R* were obtained under conditions close to equilibrium. In the region between transitions, the heat capacity decreases significantly with increasing temperature. F. Gronvold notes that for copper sulfide, for virtually every temperature, a thermodynamically equilibrium arrangement of copper atoms is formed in various voids of the crystal structure.

In work [36], phase transitions and thermal effects of solid Na_x_Cu_2-x_S samples were investigated by DSC in an argon atmosphere in the temperature range (30–430 °C). DSC was detected at 103 °C with an enthalpy area of 5.234 μW mg^−1^. The beginning of the effect is about 69 ± 3 °C. The end of the thermal effect is about 122 °C. The heat capacity varies within 150–480 J kg^−1^ K^−1^. In the series of compositions Na_0.15_Cu_1.85_S, Na_0.17_Cu_1.80_S and Na_0.20_Cu_1.77_S, prepared by an exchange reaction in a melt mixture of sodium and potassium hydroxides and cold pressing, the heat capacity at the point of phase transition increases with increasing sodium content. This can be interpreted as an indirect evidence of sodium participation in the formation of the crystal lattice of copper sulfide, as with increasing sodium concentration, energy expenditures on mixing of cations (copper and sodium) should increase. Enthalpy of the transition increases with increasing sodium content in the Na_0.15_Cu_1.85_S, Na_0.17_Cu_1.80_S, and Na_0.20_Cu_1.77_S chain (5234, 6923, and 11,720 J kg^−1^K^−1^, respectively). Extremely low thermal conductivities 0.3–0.1 W m^−1^ K^−1^ were obtained for Na_0.15_Cu_1.85_S composition. The *ZT* = 0.28 value achieved for the alloys studied is good at 320 °C for copper sulfides, but it can be greatly increased, in our view, by lowering the sodium content in non-stoichiometric copper sulfide (Cu_1.75_S ÷ Cu_1.85_S) to the optimum so that the electrical conductivity remains high enough.

In following work [39], X-ray phase analysis showed that at 20 °C, the samples Na_0.3_Cu_1.6_S, Na_0.35_Cu_1.5_S, and Na_0.4_Cu_1.55_S are a mixture of phases: monoclinic jarleite Cu_1.93_S, rhombohedral digenite Cu_1.8_S, and hexagonal Na_2_S_2_. Synthesized mixtures were compacted by cool pressing. The crystallite sizes obtained from the analysis of X-ray were 32–67 nm, 41–96 nm, and 15–37 nm for Na_0.3_Cu_1.6_S, Na_0.35_Cu_1.5_S, and Na_0.4_Cu_1.55_S samples, respectively. For all samples, there is a sharp increase in the Seebeck coefficient after 250 °C. The energy dependence of the transparency of additional barriers for current carriers arising due to the presence of inclusions of the dielectric phase (Na_2_S_2_), the content of which is proportional to the sodium concentration in the sample, can lead to an increase in the Seebeck coefficient. For the Na_0.4_Cu_1.55_S sample, a high value of *ZT* = 0.84 was obtained at 358 °C. Above 350 °C, the conductivity of all samples sharply decreases and the Seebeck coefficient rises abruptly, since the fusion of individual phases with the formation of a new phase on the basis of the cubic modification of copper sulfide (chalcocite) is possible, which is characterized by low electronic conductivity at the level of units S cm^−1^ [46,122].

In [138], the results of thermal conductivity in Cu_1.8_S compounds with the addition of Na_2_S are shown. It was found that after the introduction of Na_2_S into the Cu_1.8_S matrix, the thermal conductivity values decreased. Y. Zhang et al. [138] explain this behavior of thermal conductivity by enhanced phonon scattering due to an increase in the number of micropores, which have a great influence on the decrease in thermal conductivity; in addition, the acoustic velocity has decreased, which is considered as an effective thermal barrier, and, as a consequence, leads to a decrease in thermal conductivity.

#### 3.2.3. Possibilities of Practical Application of Copper Chalcogenides and Its Alloys

After the sobering article by Dennler [18] with doubts about the usefulness of thermoelements unstable to degradation based on copper chalcogenides, the boom of studies of thermoelectric properties of these materials did not stop, and the number of studies did not decrease. Against the background of a further gradual increase in *ZT*, we would like to note one message [224]. The message concerns the technical possibility of avoiding the release of copper from the chalcogenide during the operation of the thermoelement. In the work [224] by Qiu P. et al., through systematically investigating electromigration in copper sulfide/selenide thermoelectric materials, the mechanism for atom migration and deposition based on a critical chemical potential difference was revealed. The authors have shown that the release of copper from the sample begins when the electrical potential difference applied to the sample exceeds the critical voltage determined by equality
(31)$$V_{c}=-\frac{1}{FZ_{e}}\Delta \mu_{Cu}^{crit}-S^{*}\Delta T,$$
where *Z_e_* defines the charge (−1 for electrons or +1 for holes), *F* is Faraday’s constant, $\Delta \mu_{Cu}^{crit}$ is some critical chemical potential, and *S** accounts for the net effect of thermodiffusion. From this analysis, it is expected that a voltage difference, not current density, is the critical parameter for Cu deposition. The microscopic defect model of Yokota and Korte [225,226] allows one to determine the critical chemical potential by using the term off-stoichiometry (nonstoichiometry degree), *δ*, (in Cu_2−*δ*_X, X = S, Se) and to calculate *V_c_* as:(32)$$V_{c}=-\frac{RT}{F}\;\left[Arsinh\left(\frac{\delta_{C}}{2\sqrt{K_{e}}}\right)-Arsinh\left(\frac{2\delta -\delta_{C}}{2\sqrt{K_{e}}}\right)\right].$$

A parameter named the critical off-stoichiometry *(δ_c_)* is introduced here, corresponding to the “solubility limit” of Cu concentration at the cathode of the cell with the mixed ionic–electronic conductor. *K_e_* is the equilibrium constant for electrons and holes that is independent of stoichiometry, *R* is the gas constant, and *T* is the temperature. The thermodynamic theory developed in the work predicts that a given off-stoichiometry and temperature difference will result in limitations on the electrical potential difference that is stable across the material.

On basis of the analysis of copper release, a strategy for stable use is proposed by the authors of paper [224]: constructing a series of electronically conducting, but ion-blocking, barriers to reset the chemical potential of such conductors to keep it below the threshold for decomposition, even if it is used with high electric currents and/or large temperature differences. This strategy opens the possibility of using such conductors in thermoelectric applications and may also provide approaches to engineer perovskite photovoltaic materials.

Despite the importance of studying the physical side of the phenomena and progress in synthesis, the practical application of copper sulfide and its alloys is no less important. Today, materials with *ZT* ~2 and higher are considered promising for use on an industrial scale. In addition to this main criterion for the development of thermoelectric devices, important conditions are cheapness and availability of raw materials, ease of synthesis, material stability and compatibility with other components of the thermoelectric module, and others [4,82,83]. Doping with lithium and sodium makes it possible to enhance the useful properties of these materials, in particular, to increase the thermoelectric effect, and the creation of composites with nanoscale carbon blocks and numerous pores makes it possible to achieve thermal conductivity up to 0.1 W m^−1^ K^−1^.

## 4. Conclusions and Suggestions

Copper chalcogenides and their alloys are promising thermoelectric materials that have recently attracted increased attention of researchers. They clearly show the interaction of the electronic, ionic, and phonon subsystems of the crystal [1,20,144]. Thus, the “liquid-like” cationic sublattice is responsible for very low thermal conductivity, while nonstoichiometric defects provide high electronic conductivity, which contributes to the high thermoelectric figure of merit of copper chalcogenides [1]. Of course, high mobility of copper ions creates a problem of copper release in a long operation at high temperatures; however, this problem is surmountable. On the basis of the analysis of copper release, a strategy for stable use was proposed by P. Qiu et al. [224]: constructing a series of electronically conducting, but ion-blocking, barriers to reset the chemical potential of such conductors to keep it below the threshold for decomposition, even if it is used with high electric currents and/or large temperature differences. This strategy opens the possibility of using such conductors in thermoelectric applications.

Meanwhile, the *ZT* of alloys based on copper chalcogenides approached the value 3. Thus, *ZT* = 2.62 was obtained for the composition Cu_1.94_Al_0.02_Se at a temperature of 756 °C by B. Ghong et al. [177] and *ZT* = 2.63 was achieved for Cu_2_Se + 1 mol% CuInSe_2_ at 577 °C by A. Olvera et al. [85]. Doping, as well as the improvement of methods and conditions of synthesis, remain the main tools of researchers.

Doping of copper chalcogenides with lithium is convenient due to the proximity of the ionic radii of copper and lithium, which favors the formation of solid solutions. For the composition Li_0.09_Cu_1.9_Se, the maximum *ZT* ≈ 1.4 at 727 °C was obtained [30]. Due to the large difference in the ionic radii of copper and sodium, the solid solubility of sodium does not exceed a few percentage points, but this turned out to be enough to reach *ZT* = 2.1 at 700 °C for microporous Na_0.04_Cu_1.96_Se obtained by hydrothermal method and hot pressing [53]. It is possible that the result would have been higher if a more non-stoichiometric composition was chosen as the matrix for doping with sodium, e.g., Cu_1.94_Se. Alloys of copper chalcogenides with high sodium content are not homogeneous and form a mixture of phases with different conducting and thermoelectric properties, but deserve attention as nanocomposite thermoelectric materials with low thermal conductivity. The properties of such mixtures are poorly understood, but we expect good thermoelectric performance here associated with the presence of numerous barrier layers and pores.

Not only superionic conductors have low lattice thermal conductivity, but also other classes of materials, such as multi-filled skutterudites [117], for example. The multiple-filled skutterudite TE material Ba*_u_*La*_ν_*Yb*_w_*Co_4_Sb_12_ prepared by spark plasma sintering [7] shows *ZT* ~1.7 at 577 °C [115]. The very high *ZT* value benefits from the following two aspects: (a) controlling the filling fraction of multiple filled atoms to optimize the carrier concentration, leading to a higher power factor and (b) strong scattering of wide-band phonons can be achieved by different rattling frequencies of the multiple atomic filling in the icosahedral voids of CoSb_3_ TE material, making the lattice thermal conductivity close to the theoretical minimum.

The results of recent years show that the decrease in thermal conductivity due to nanostructuring and nanoinclusions of another phase is comparable in magnitude with the contribution of “melting” of the sublattice of mobile ions in a super-ionic conductor.

Moreover, do not underestimate heterogeneous thermoelectrics and composites. The recent works reveal that inclusions of the second phase, such as the addition of graphene or carbon fiber, make it possible to achieve a sharp increase in *ZT*. Thus, L. Zhao et al. [178] achieved *ZT* = 2.4 for composition Cu_2_Se + 0.3 wt.% carbon fiber at 577 °C.

In our opinion: bulk materials soon will be able to exceed *ZT* = 3. Recently, the highest *ZT* = 2.8 at 500 °C was achieved for SnSe_1−x_Br_x_ composition by C. Chang et al. [91], who maintained that a continuous phase transition increases the symmetry and diverges two converged conduction bands. These two factors improve carrier mobility, while preserving a large Seebeck coefficient.

## Figures and Tables

**Figure 1 nanomaterials-11-02238-f001:**
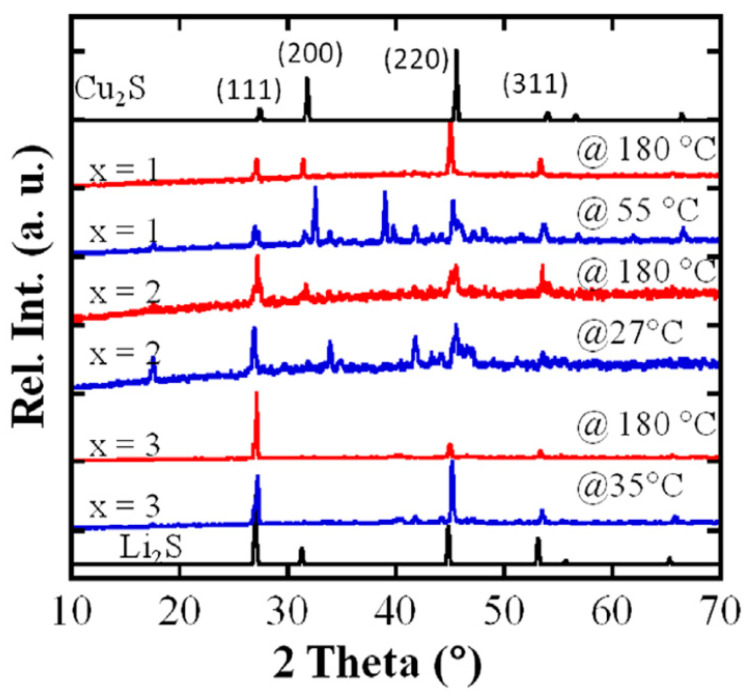
Temperature dependences of the X-ray patterns of the Cu_4−*x*_Li*_x_*S_2_ series powders in comparison with the calculated diagrams of the Cu_2_S and Li_2_S powders. Reprinted with permission from ref. [77]. Copyright 2015 Elsevier.

**Figure 2 nanomaterials-11-02238-f002:**
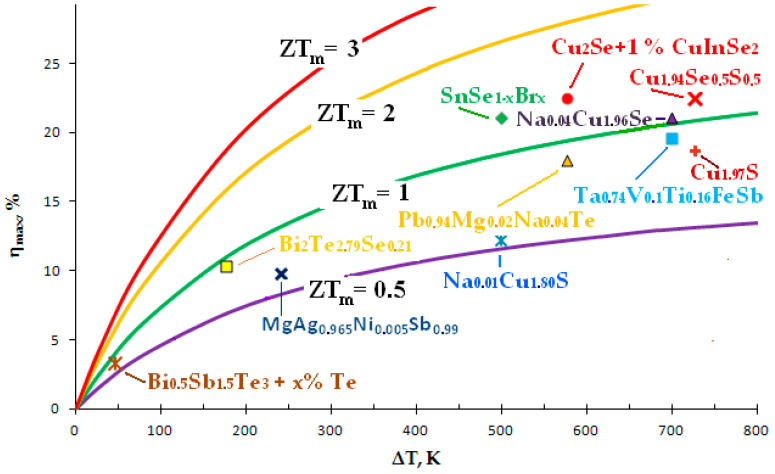
Thermoelectric efficiency of some thermoelectric materials: Cu_1.97_S [84], Na_0.04_Cu_1.96_Se [53], Na_0.01_Cu_1.80_S [31], Cu_2_Se + 1% CuInSe_2_ [85], Cu_1.94_Se_0.5_S_0.5_ [86], Bi_0.5_Sb_1.5_Te_3_ + x% Te [87], Ta_0.74_V_0.1_Ti_0.16_FeSb [88], Pb_0.94_Mg_0.02_Na_0.04_Te [89], MgAg_0.965_Ni_0.005_Sb_0.99_ [90], SnSe_1-x_Br_x_ [91], Bi_2_Te_2.79_Se_0.21_ [92].

**Figure 3 nanomaterials-11-02238-f003:**
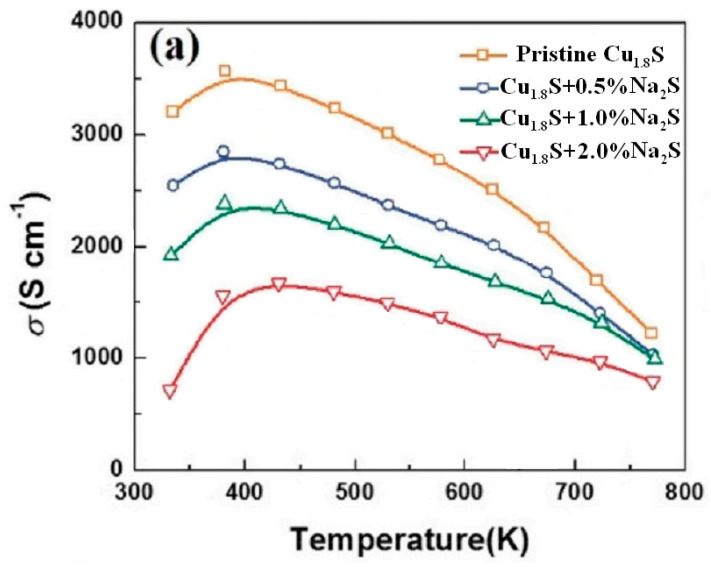
Temperature dependence of electrical conductivity of bulk Cu_1.8_S samples doped with Na_2_S (*x* = 0, 0.5, 1, 2 wt.%). Reprinted with permission from ref. [138]. Copyright 2020 Elsevier.

**Figure 4 nanomaterials-11-02238-f004:**
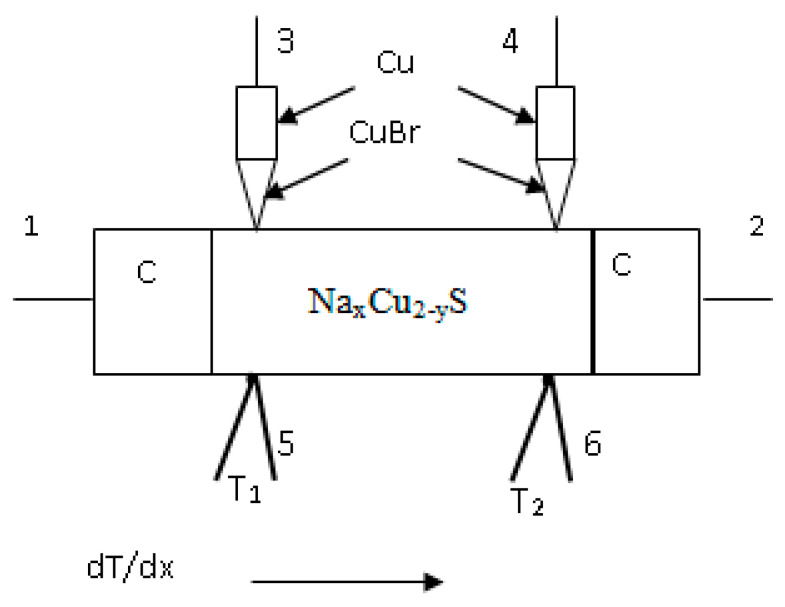
Schematic diagram of a cell for measuring the coefficients of electronic and ionic thermo-emf and electronic conductivity. In the picture, symbols C and c denote Cu probes and electrodes, CuBr is ionic conductor, Na_x_Cu_2-y_S is sample (example), T- thermocouple. The control of the constancy of the chemical composition and the equilibrium state of the samples during measurements was carried out by the EMF of the electrochemical cell Cu / CuBr / sample / Pt, measured between pairs of contacts 3-5 and 4-6, in two cross sections of the sample. Reliable clamping of potential probes 6 and thermocouples 4 to the sample was ensured using flexible wire 5, tightened by springs.

**Figure 5 nanomaterials-11-02238-f005:**
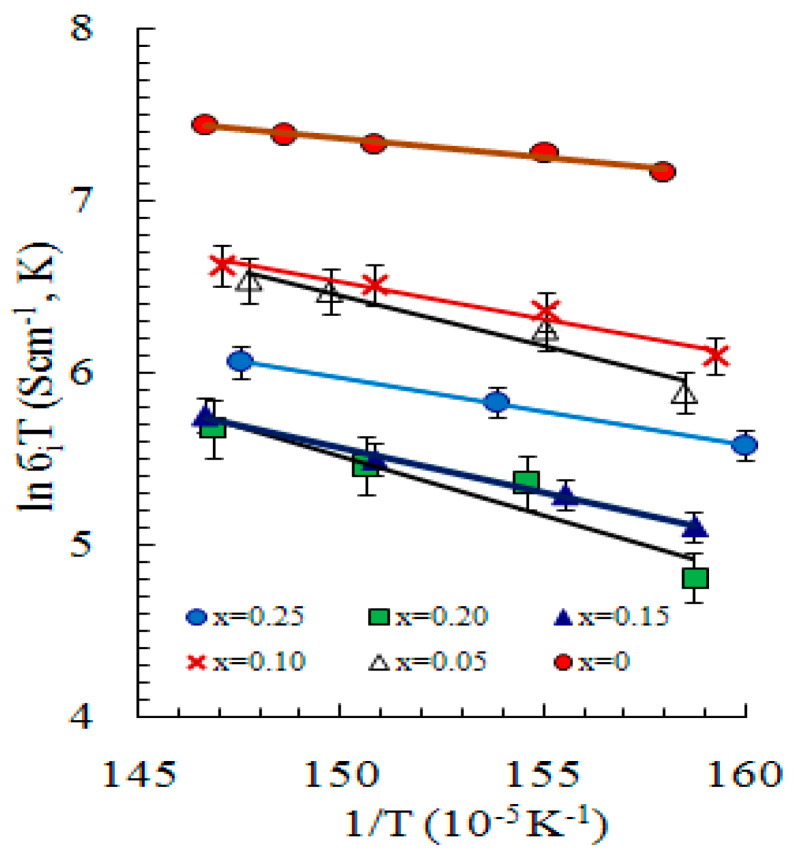
Temperature dependences of the ionic conductivity of Cu_2_S and Li*_x_*Cu_2−*x*_S samples. Reprinted from reference [24]. Copyright 2003 John Wiley and Sons.

**Table 1 nanomaterials-11-02238-t001:** Parameters of ion transport in solid solutions Li_x_Cu_2-x_S near a temperature of 355 °C. Reprinted from reference [24]. Copyright 2003 John Wiley and Sons.

Structure	*σ_i_* (S cm^−1^)	*E_a_* (eV)	$\tilde{E_{a}}$ (eV)
Cu_2_S	2.0	0.19 ± 0.02	0.23 ± 0.02
Cu_1.95_Li_0.05_S	0.57	0.50 ± 0.10	0.54 ± 0.03
Cu_1.90_Li_0.10_S	0.71	0.37 ± 0.06	0.30 ± 0.04
Cu_1.85_Li_0.15_	0.26	0.45 ± 0.04	0.51 ± 0.09
Cu_1.80_Li_0.20_S	0.20	0.59 ± 0.10	0.49 ± 0.02
Cu_1.75_Li_0.25_S	0.58	0.33 ± 0.01	0.28 ± 0.04

**Table 2 nanomaterials-11-02238-t002:** State-of-art thermoelectric properties of some perspective materials.

Materials	Year	Synthesis	*α*, mVK^−1^	*σ*, S cm^−1^	*PF* = *α*^2^*σ*, μWcm^−1^K^−2^	*k*, Wm^−1^K^−1^	*T*, °C	*ZT_max_*	Ref.
Cu_1.98_S_1/3_Se_1/3_Te_1/3_ (p-type)	2017	Melting + annealing + SPS	0.243	182	10.7	0.57	727	1.9	[179]
Cu_1.97_S (p-type)	2014	Melting + annealing + SPS	0.3	100	8.2	0.48	727	1.7	[84]
Na_0.01_Cu_1.80_S (p-type)	2016	Mechanical Alloying + SPS	0.110	850	10.5	0.7	500	1.1	[31]
Na_0.04_Cu_1.96_Se + micropores (p-type)	2019	Hydrothermal method + HP	~0.29	~130	~11	~0.54	700	2.1	[53]
Cu_1.98_Li_0.02_S + nanopores	2018	Hydrothermal method + HP	~0.27	~145	~10.6	0.48	700	2.14	[200]
Cu_1.94_Al_0.02_Se	2014	Melting + BM + SPS	0.246	261	15.8	0.611	756	2.62	[177]
Cu_2_Se + 1 mol% CuInSe_2_ (p-type)	2017	BM + SPS	0.15	550	12.4	0.4	577	2.63	[85]
Cu_1.94_Se_0.5_S_0.5_ (p-type)	2017	Melting + annealing + SPS + HP	0.37	~96	13.2	~0.6	727	2.3	[86]
Cu_2-x_S + 0.75 wt% Grapheme (p-type)	2018	BM+SPS + annealing in 95 vol% Ar and 5 vol% H_2_	~0.16	~450	~12	0.67	600	~1.5	[169]
Ag_2_Sb_0.02_Te_0.98_ (n-type)	2020	Vacuum melting + +annealing at 1000 °C and 400 °C + cooling at −173 °C	~0.106	~870	~9.8	~0.29	137	1.4	[166]
Cu_2_Se + 0.3 wt.% carbon fiber (p-type)	2017	Solid state Cu_2_Se synthesis + BM + CP + annealing	0.175	375	11.5	0.4	577	2.4	[201]
Bi_0.5_Sb_1.5_Te_3_ + x% Te (p-type)	2015	Liquid-phase compacting + SPS	~0.24	~650	~37	~0.65	47	1.86	[87]
Pb_0.940_Mg_0.020_Na_0.040_Te (p-type)	2016	Solid state synthesis from elements + SPS	~0.24	~400	~23	~1	577	1.8	[89]
PbTe + 0.2% PbI_2_ (n-type)	2016	Solid state synthesis from elements + SPS	~0.23	~330	~20	~1.2	477	1.4	[89]
Ta_0.74_V_0.1_Ti_0.16_FeSb (p-type)	2019	BM + HP	~0.225	~1040	~52	~3.35	700	1.52	[88]
MgAg_0.965_Ni_0.005_Sb_0.99_ (p-type)	2015	BM + HP	~0.235	~450	~25	~1.1	245	~1.15	[90]
SnSe_1−x_Br_x_ (n-type)	2018	The temperature gradient method and bromine doping	~0.48	~38	~9	~0.245	500	2.8	[91]
Bi_2_Te_2.79_Se_0.21_ (n-type)	2015	Zone melting + BM + HP + hot deformation	~0.192	~970	~36	~1.1	84	1.2	[92]

BM is ball milling, SPS is spark plasma sintering, HP is hot pressing, CP is cold pressing, T is temperature, *α* is Seebeck coefficient, *σ* is electrical conductivity, *k* is thermal conductivity, PF is power factor, and ZT_max_ is the peak value of figure of merit.

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
