# Peer review of "Some Thermoelectric Phenomena in Copper Chalcogenides Replaced by Lithium and Sodium Alkaline Metals"

_nanomaterials, 2021, doi:10.3390/nano11092238_

Round 1
Reviewer 1 Report
In this manuscript, the authors systematic reviewed the thermoelectric properties of copper chalcogenides substituted with sodium and lithium alkali metals, especially the Na-Cu-S and Li-Cu-S ternary systems. The preparation methods and their electrical, thermodynamic, thermal, thermoelectric properties of nanocrystalline copper sulfides and its alloys also have been discussed. Overall, this work is suitable for publication on Nanomaterials in my opinion. In the meantime, some comments and suggestions should be carefully revised before acceptance:
- In table 1, the second row filled by numbers of 1 to 9 with no meaning and it should be deleted.
- The format of “℃” in the manuscript is inconsistency, the authors should be carefully double check. Besides, the authors describe temperature with two unites as “℃” and “K”, it should be better to have a unified unit.
- There is no table 2 in the manuscript, please give the right serial number of all tables.
- For the references, the format also should be double checked, especially the abbreviation of journal names and the number of pages.
Author Response
Dear Reviewer,
the authors are very grateful to you for your fair and useful comments. We tried to follow them and made the necessary changes to the article.
In this manuscript, the authors systematic reviewed the thermoelectric properties of copper chalcogenides substituted with sodium and lithium alkali metals, especially the Na-Cu-S and Li-Cu-S ternary systems. The preparation methods and their electrical, thermodynamic, thermal, thermoelectric properties of nanocrystalline copper sulfides and its alloys also have been discussed. Overall, this work is suitable for publication on Nanomaterials in my opinion. In the meantime, some comments and suggestions should be carefully revised before acceptance:
- In table 1, the second row filled by numbers of 1 to 9 with no meaning and it should be deleted.
We agree. This line has been removed from Table 1
- The format of “℃” in the manuscript is inconsistency, the authors should be carefully double check. Besides, the authors describe temperature with two unites as “℃” and “K”, it should be better to have a unified unit.
We agree. Temperature is described in “ ℃” scale across the article.
- There is no table 2 in the manuscript, please give the right serial number of all tables.
Yes. The corrections are made for table numbers.
- For the references, the format also should be double checked, especially the abbreviation of journal names and the number of pages.
Ok. All references are checked and corrected.
Thank you very much!
Please see the attachment.

Reviewer 2 Report
This paper by Kubenova et al. reviewed recent studies on thermoelectric transport in alkali metal-substituted copper chalcogenides. The paper is well-written and can be a good contribution to the field. I only have minor comments before it can be accepted for publication.
1. While described in short in Conclusions, it will make the review more interesting if the authors can describe the perspective and challenges in the field in more detail as an independent section.
2. There are typological errors such as “zT” and “ZT” on page 33, “Ea” on page 27 should be lower-cased, and symbols are arbitrarily italicized or non-ilaticized throughout the manuscript.
3. Styles of the references need to be checked carefully. For instance, there are many errors in the journal abbreviations.
Author Response
Dear Reviewer,
the authors are very grateful to you for your fair and useful comments. We tried to follow them and made the necessary changes in the article.
This paper by Kubenova et al. reviewed recent studies on thermoelectric transport in alkali metal-substituted copper chalcogenides. The paper is well-written and can be a good contribution to the field. I only have minor comments before it can be accepted for publication.
- While described in short in Conclusions, it will make the review more interesting if the authors can describe the perspective and challenges in the field in more detail as an independent section.
We tried to take into account your remark and added a few sentences in the conclusion regarding the prospects and challenges in the field.
- There are typological errors such as “zT” and “ZT” on page 33, “Ea” on page 27 should be lower-cased, and symbols are arbitrarily italicized or non-ilaticized throughout the manuscript.
We agree. The corrections are made across the article.
- Styles of the references need to be checked carefully. For instance, there are many errors in the journal abbreviations.
Ok. All references are checked and corrected.
Thank you very much!
Please see the attachment.

Reviewer 3 Report
The present paper is considered as a review paper, on Li and Na doped Copped Chalcogenides, and their thermoelectric properties. Unfortunately, the paper is not strong enough for publication, for the following reasons:
- Introduction: Although detailed, the introduction lacks of a general purpose, which will be given to the reader and the latter will clearly understand the scope of the paper.
- Line 218: Is this a header or an incomplete sentence?
- Equations: There are a lot of equations describing basic solid state physics concepts. I do not believe that they contribute to the paper, as such.
- Figures and pictures must be improved, regarding their resolution.
- Line 1117: “Thermoelectric properties and thermal conductivity”, Such title is not very clear to me. Could it be changed to “Seebeck coefficient and thermal conductivity”?
- There are results presented, regarding not only Na and Li doped chalcogenides, but other thermoelectric materials (i.e., table 4). That does not follow the title of the paper.
- Conclusion part is weak. The sentence “Currently, insufficient attention is paid to heterogeneous materials and composites” cannot be true since there are plenty of research papers on thermoelectric composites, while lots of research groups are dedicated in such research field. Furthermore, the opinion “In our opinion, it is the heterogeneous bulk materials that will make it possible to exceed the threshold ZT = 3, the achievement of which is a task for the near future.” is not actually supported. Why is this “the task for the near future”?
Therefore, I propose the authors consider the above comments, and correspondingly revise the manuscript.
Author Response
Dear Reviewer,
the authors are very grateful to you for your useful comments. We tried to follow them and correspondingly revised the manuscript.
The present paper is considered as a review paper, on Li and Na doped Copped Chalcogenides, and their thermoelectric properties. Unfortunately, the paper is not strong enough for publication, for the following reasons:
- Introduction: Although detailed, the introduction lacks of a general purpose, which will be given to the reader and the latter will clearly understand the scope of the paper.
Yes, we are agree. In the introduction, a fragment of the following text has been added:
The general goal of this review is to consider copper chalcogenides and their alloys as thermoelectric materials from the point of view of the researcher of the superionic state, since it is this state that provides these materials with beautiful thermoelectric properties and excellently low thermal conductivity. In addition, we wanted to show convenient electrochemical methods for working with copper and silver chalcogenides, which are not in demand by the current generation of researchers. These ideas developed in the works of C. Wagner, I. Yokota, S. Miyatani, for example, the coulometric titration method and methods for measuring transport characteristics in direct dependence on Fermi level are very suitable for tuning of thermoelectric power by change of the copper content. However, it is the superionic state that complicates the practical application of copper and silver chalcogenides, since the mobility of cations similar to the mobility of atoms in a liquid leads to the release of the metal from the sample during a long operation at high temperatures. Therefore, we considered here also the possibilities of overcoming this important problem: associated with a decreasing in ionic conductivity of the material and with the design of thermoelectric modules, developed taking into account the threshold of metal release from chalcogenide.
In addition, we consider here some other TE materials, such as half-Heusler phases, skutterudites, etc., so that the reader has the opportunity to compare and can more fully evaluate the superionic copper chalcogenides as thermoelectric materials.
- Line 218: Is this a header or an incomplete sentence?
Unfortunetely, we do not have the same line numbering.
Perhaps the line "Solid solutions on base of copper sulfide" was meant. This is a level four heading.
- Equations: There are a lot of equations describing basic solid state physics concepts. I do not believe that they contribute to the paper, as such.
Perhaps it was unnecessary, but in the second version of the article, we added several equations for semiconductor systems, since the Reviewer â„– 2 noticed that our consideration is more suitable for metallic systems, while thermoelectrics are mostly semiconductors.
- Figures and pictures must be improved, regarding their resolution.
The figures have been improved as much as possible, as many of the figures are from other articles.
- Line 1117: “Thermoelectric properties and thermal conductivity”, Such title is not very clear to me. Could it be changed to “Seebeck coefficient and thermal conductivity”?
Ok, we changed the title of the section to your suggested one.
- There are results presented, regarding not only Na and Li doped chalcogenides, but other thermoelectric materials (i.e., table 4). That does not follow the title of the paper.
We made it due to the reader has the opportunity to compare and can more fully evaluate the superionic copper chalcogenides as thermoelectric materials.
- Conclusion part is weak. The sentence “Currently, insufficient attention is paid to heterogeneous materials and composites” cannot be true since there are plenty of research papers on thermoelectric composites, while lots of research groups are dedicated in such research field.
The sentence “Currently, insufficient attention is paid to heterogeneous materials and composites” is deleted from the manuscript.
Furthermore, the opinion “In our opinion, it is the heterogeneous bulk materials that will make it possible to exceed the threshold ZT = 3, the achievement of which is a task for the near future.” is not actually supported. Why is this “the task for the near future”?
Conclusion part is revised.
Therefore, I propose the authors consider the above comments, and correspondingly revise the manuscript.
Thank you very much!
Please see the attachment.

Round 2
Reviewer 3 Report
After authors' revisions the paper became suitable for publication.
This manuscript is a resubmission of an earlier submission. The following is a list of the peer review reports and author responses from that submission.
Round 1
Reviewer 1 Report
In this manuscript, the authors report on a comprehensive review of the crystal structure and physical properties of copper chalcogenides, with a particular emphasis on their thermoelectric properties. These compounds have attracted strong interest in the TE community due to the report of high ZT values. However, these compounds suffer from their propensity to show ionic conduction, detrimental to their use in thermoelectric generators. The present authors do not "hide" this drawback and took it fully into account. For these reasons, I would recommend publication of this review in Nanomaterials. Prior to acceptance, I would suggest the following points to be addressed to further improve the manuscript:
1). I would recommend to show crystal structures drawn with a dedicated crystallographic software (such as VESTA which is free) for Figures 1 and 2. Moreover, Figure 3 appears to be blurred and is thus not of high quality.
2). In Table 1, the word "persecution" is probably a typo.
3). The different formulas used to describe the transport in Section 2.3.1 are those derived for metallic systems. They are not necessarily useful for semiconducting compositions. Perhaps the authors may add some comments about transport models (single-parabolic band, non-parabolic...) used to describe them in this case.
4). In the section dedicated to the thermal properties, some results obtained by inelastic neutron scattering on related compounds may be worth being quoted and discussed.
5). As a final part, I would recommend adding a conclusion and perspectives paragraph summarizing the current state of the field and possible future research directions.
Reviewer 2 Report
Recommendation: major revisions needed as noted
Comments:
The uuthors reported the review of “Some Thermoelectric Phenomena in Copper Chalcogenides Replaced by Lithium and Sodium Alkaline Metals”, it’s quite interesting to read the copper chalcogenides replaced by lithium and sodium alkaline metals. However, there are several issues that should be addressed before this manuscript can be published.
- Page number 1, line 18, Keywords are missing, please add minimum 5 key words.
- Page number 1, Line 23, In general, thermoelectric materials are defined as TE materials instead of TEM. It is advised to use TE materials. Because TEM looks like Transmission Electron Microscopy (TEM)
- In the section of introduction, line 36, In equation (1) please use conventional notation for thermal conductivity kappa (k) and the unit of electrical conductivity is S/m or Ohm-1 m-1.
- Page number 2, line 86 and 89, use the same format like 1.4 at 1000 K instead of 1.4 – at 1000 K
- Page number 3, line 96, 108, 110, 119, 120 etc., use 103 °C. Should give some space between number and unit in whole manuscript, tables and references.
- Page number 3, line 131, In the section of introduction, the authors made very strong broad statement "The CuS phase, which is present in some samples, is stable up to 507 °C, then transforming into the cubic Fm3m phase. Thus, when heated, the material becomes homogeneous, although at room temperature it is multiphase", please list supporting references.
- Page number 3, line 138, the authora gave figure 1 at the same time page number 7 line 308 author type Fig.2. use fig or figure, follow the same standard MDPI nanomaterials format.
- Page number 3 In Table (1), units of lattice parameters are missing.
- Should check sample names subscript and superscript in the whole manuscript, including all the tables and references, for example and sample name not InNi2, CaF5 should be in this format InNi2, CaF5.
- In table 1, 3-digit or four-digit number, it’s very confusing, please check the typos.
- In table 1, Lattice parameters are often reported as a, b, c but you have reported a, b, c and a, b, e so, please explain why you have reported a, b, c and a, b, e?
- Give the comparison table for thermoelectric properties and synthesis method of copper chalcogenides.
- Page number 6, line 219, 244, reference are missing P. Qiu et al. [ ], G. Denler et al [ ]. Page number 8, line 334 Kurt O.et al [ ] ref missing. Please check the whole manuscript properly and add the ref.
- Page number 9 line 358, The authors of [93] investigated a semiconductor alloy with the composition Li15Cu1.85S with an ionic conductivity (8-10) times lower than that of pure copper sulfide, characterized in that it was obtained by pressing from Li0.15Cu1.85S nanopowder. What kind of pressing they have used, hot pressing or cold pressing? Please give more detailed in this statement if possible.
- Page number 9 line 368, the authors wrote: theoretical calculations of the band gap for LiCuS (1.72) by the Engeland Vosko method, what is the lattice parameters of LiCuS?
- Page number 9 line 371, the authors wrote: The TGA curves shows sharp endothermic peaks around 410 K, reflecting a phase transition, please provide TGA and XRD figure and line line 375 author says decreasing the thermal conductivity with an increasing in temperature and Cu:Li ratio, please give some refs and if possible provide graph also.
- Page number 12, method 2.2, In methods for the synthesis of nanocrystalline copper sulfide and its alloys, authors explain the synthesis of others compounds like MgAgSb, ZrCoBi. Only few synthesis methods of copper chalcogenides are explained, the morphology and solidification method are not explained Please also explain the advantages of synthesis method for thermoelectric properties.
- Page number 14, line 563, the authors wrote: ni is also significantly less then nt”. Is it ni represents Ni? What about nt? Please provide clear abbreviation.
- Page number 15, line 591, the authors wrote: ne and µn are the concentration and mobility of electrons, please correct them to ne, µn (e and n subscript).
- Page number 18, line 732, the authors wrote: a X. Zhang et al [82] etc., Pauli paramagnetism with a value of wm 6.2 ´ 10-5 cm/mol-1, but from the literature X. Zhang et al. have reported a Pauli paramagnetism with a value of cm = 6.2 ´ 10-5 emu mol-1. Please check the unit.
- Page number 19 line 744, the authors wrote: as shown in equation (7) the Na+ ion enters the Cu9S5 I can’t see any Na+ and Cu9S5 lattice in the equation 7. Please check Change equation number
- Page number 19, line 758, the authors wrote: Na doping increase the resistivity from 53 to 77 10-6 ohm * m. Is it 53 to 77 ´ 10-6 ohm * m?
- Page number 19, line 762, the authors wrote: an electronic conductivity of Na2S (X=0, 0.5, 1, 2 wt.%). It should be Na2Sx instead of Na2
- Page number 20 table 2, Should write electrical conductivity like text in table. The authors wrote Om -1 cm-1, please check the typo.
- Page number 24, line 949, the authors wrote: a cu is the heat capacity of 1 CM3 of the crystal. Please use C please and units cm3.
- Page number 27, line 1079, the authors wrote: Sodium ion batteries (NIA), Should NIA be changed to NIB, Sodium-ion battery (NIB).
- Page number 28, line 1125, the authors wrote: (002) lattice spacing is increased to 1.2 nm but in the figure 11 shows the lattice spacing for (002) plane 1.3 nm. It is a typo?
- Please add a section of summary.
- I suggested to change the part 2.4, “Thermal and thermoelectric properties of copper sulphide and its alloy” into “Thermoelectric properties and thermal conductivity of copper sulphide and its alloy”. Author should discuss TE properties first and then thermal conductivity, it would be easier to understand the materials.
- Overall, this manuscript described materials and synthesis techniques, then the authors provided less information about thermoelectric properties in this manuscript.
- There are some typos. Therefore, the authors are advised to recheck the whole manuscript for improving the language carefully.

Reviewer 3 Report
The authors review the current status of thermoelectric (TE) research on Cu chalcogenides, with a particular focus on Li/Na replacement as a mean to improve the material stability and TE performance.
I am afraid the review is basically not ready for review and strong revisions are necessary before acceptance. Few points in need of attention:
>> A general problem is the following: in a review, I would expect some additional elaboration on the key ideas and concepts. I find the paper is almost exclusively (maybe I would partially save the intro, which is more pertinent to a review) a long stream of specific examples, without much elaboration or even with no elaboration at all. I also miss a more precise comparison with TE materials in general. I just see a graph in Fig.5, way into the review (I would expect some discussion at the very beginning), but no copper-based material appears in the graph.
>> The quality of the presentation is below par in various parts. Few examples:
- Destructured paragraphs. Consider lines 104-136: we have a long "dump" of single-sentence paragraphs. Again I would expect some elaboration or, if the authors want to make an unstructured list of examples... well, then they should make a (explicit) list of examples: one bullet for each item etc. Similar situation elsewhere, for instance consider lines 349-369: 8 paragraphs is 30 lines. Or consider line 532 (a further random example among many): a single-sentence paragraph, rather loosely connected to the text above and below, stating "M.R.Gao [124] presents more than 15 liquid-phase methods for the synthesis and modification of chalcogenide nanomaterials": this is good to know, but I don't think this statement is very informative... either the authors explain what all this is about and why is it interesting or this is equivalent (but less readable) than putting reference 124 in a long list of papers about "the many ways we have to create Cu-based nanomaterials [... 124, ...]."
- At some point the authors start to talk about batteries (around 418). Is this relevant? This seems out of scope to me.
- I suggest a thorough check-up of references. I just checked few and there are clearly errors. For instance Snyder is cited many times as [1], but the first reference is a different one. I guess they authors wanted to rather cite [7] maybe?
- General re-read is necessary. Few examples of things that should disappear:
* Anilite is define as $Cu_{1.75}S$ and as $Cu_7S_4$ in the line immediately below. Ok, 7/4 = 1.75 and so this is equivalent and it's a minor problem, but - for the sake of the reader - please be consistent.
* Missing sentences, words. A random sampling: line 137 space group is missing; 415 "conductiviy of the alloy ..."; etc
* Un/barely-defined symbols: ni at 390 (and in many further spots) should not appear in roman (italic, and "i" should be a pedex) and should be properly defined; what is "nt" immediately after? later on we learn it is probably the density of traps, but these are not "universal" symbols and should be properly defined; "emf" is used in 415, defined at 996, but then the acronym is not really used; what is \theta at line 962; line 814 SIT underfined; etc.
>> I am missing something even remotely resembling a conclusion. This is consistent with the general lack of "structure" of the work. I don't think a review on the TE phenomena in Cu-calcogenides can end with the sentence starting at line 1133, talking about the drawbacks of Na-based systems for building good electrodes for batteries. What is the link with the subject of the paper that I expect based on the title? Indeed somehow the final "application" section 2.4.3 at line 1076 suddenly "flips" to discussing about batteries and in the end some 80% of 2.4.3 is actually about batteries. What happened here? Unclear. This almost looks like a copy-paste from an intro of an unrelated paper talking about something else. [By the way, flipping back to the final sentence, about (but not exactly) the same sentence is reported also at the end of the previous paragraph, starting from line 1126. So I reiterate here: please re-read this paper with care].